# Multiple Bioimaging Applications Based on the Excellent Properties of Nanodiamond: A Review

**DOI:** 10.3390/molecules28104063

**Published:** 2023-05-12

**Authors:** Xinyue Wang, Dandan Sang, Liangrui Zou, Shunhao Ge, Yu Yao, Jianchao Fan, Qinglin Wang

**Affiliations:** 1Shandong Key Laboratory of Optical Communication Science and Technology, School of Physics Science and Information Technology, Liaocheng University, Liaocheng 252000, China; lcuwangxinyue@163.com (X.W.); zouliangruilcu@163.com (L.Z.); geshunhao223@163.com (S.G.); lcuyaoyu0814@163.com (Y.Y.); 2Shandong Liaocheng Laixin Powder Materials Science and Technology Co., Ltd., Liaocheng 252000, China; 13676388216@126.com

**Keywords:** nanodiamonds, bioimaging, fluorescence imaging, Raman imaging, X-ray imaging, magnetic modulation fluorescence imaging, magnetic resonance imaging, cathodoluminescence imaging, optical coherence tomography imaging

## Abstract

Nanodiamonds (NDs) are emerging as a promising candidate for multimodal bioimaging on account of their optical and spectroscopic properties. NDs are extensively utilized for bioimaging probes due to their defects and admixtures in their crystal lattice. There are many optically active defects presented in NDs called color centers, which are highly photostable, extremely sensitive to bioimaging, and capable of electron leap in the forbidden band; further, they absorb or emit light when leaping, enabling the nanodiamond to fluoresce. Fluorescent imaging plays a significant role in bioscience research, but traditional fluorescent dyes have some drawbacks in physical, optical and toxicity aspects. As a novel fluorescent labeling tool, NDs have become the focus of research in the field of biomarkers in recent years because of their various irreplaceable advantages. This review primarily focuses on the recent application progress of nanodiamonds in the field of bioimaging. In this paper, we will summarize the progress of ND research from the following aspects (including fluorescence imaging, Raman imaging, X-ray imaging, magnetic modulation fluorescence imaging, magnetic resonance imaging, cathodoluminescence imaging, and optical coherence tomography imaging) and expect to supply an outlook contribution for future nanodiamond exploration in bioimaging.

## 1. Introduction

Nanodiamonds (NDs) are considered a potentially very promising material for multifunctional applications in biomedical studies due to their extensive confirmed biocompatible applications. Particularly, the optical and spectral properties of NDs have been considered for different methods of multimodal bioimaging (including drug delivery tracing) based on ND fluorescence [1,2]. The fluorescence of NDs is attributed to the existence of optical defects inside the diamonds, among which are defects that can make electrons leap in the forbidden band and absorb or emit light during leaping, called color centers, which can selectively absorb visible light energy and emit fluorescence. The color centers in nanodiamonds are nitrogen vacancy (NV), silicon vacancy (SiV), nickel vacancy and chromium-related color centers, among which NV and SiV are the most common. NV centers are formed by replacing a carbon vacancy and the adjacent carbon atom with a nitrogen atom in the face-centered cubic diamond lattice [3], while SiV centers are formed in a similar way to NV centers, replacing a carbon vacancy and the adjacent carbon atom with a silicon atom in the nanodiamond lattice [4]. The emission spectra of NDs’ NV center reflects the interactions on the nanodiamond surface and the number of NV centers inside the particles [3]. NV centers exist mainly in neutral and negative charge states, and their broad-spectrum stable emission at room temperature makes them widely used in bioimaging [5]. However, the excitation wavelength of NV centers is usually in the range of 510–560 nm, making it easily absorbed by organic molecules, causing problems of tissue autofluorescence interference and low tissue penetration depth. In this case, nanodiamond can be made to fluoresce using SiV centers instead of NV centers because the excitation wavelength of SiV centers is usually around 885 nm, which can overcome the problem of the autofluorescence of biological tissues. The research on the optical defects of nanodiamonds lays the foundation for future applications of nanodiamonds in bio-diagnostics and biomedicine. Nanodiamonds are inert and chemically stable, which makes them fairly non-reactive with other substances, and therefore suitable as a bioimaging probes.

There are various types of NDs, and we can classify NDs using different criteria. Firstly, according to the spatial size, they can be classified into zero-dimensional ND single crystal particles, one-dimensional diamond nanorods, two-dimensional diamond nanosheets, three-dimensional ND polycrystalline particles, and ND films. Secondly, they can be classified according to the synthesis methods of NDs, which can be divided into detonation nanodiamonds (DNDs), high pressure and high temperature nanodiamonds (HPHT NDs), high energy ball milling NDs, laser ablation NDs, and chemical vapor deposition NDs [6]. DNDs are synthesized by the carbon of an explosive system, which is transformed into diamond by the high temperature and pressure generated by explosive mixtures detonated in the absence of oxygen. The synthesized NDs surfaces have many surface groups (including oxygen-containing functional groups or other functional groups) that change the surface potential of the DNDs, causing the DNDs to agglomerate, resulting in generally aggregated NDs [6]. Generally, the size of DNDs after isolation and purification is 4–5 nm. DNDs are highly biocompatible and their numerous surface groups make them easy to surface modify, which is beneficial for bioscience applications. HPHT NDs are synthesized by transforming graphite powder into NDs under quasi-hydrostatic pressure and high temperature. In recent studies, naphthalene (C_10_H_8_ (Chemapol)) and organosilicon compounds such as tetrakis (trimethylsilyl) silane (C_12_H_36_Si_5_ (Stream Chemicals)) were generally used as precursors and then the high-temperature and high-pressure methods were used to synthesize HPHT NDs [4,6]. In a new study, HPHT NDs were synthesized with a size of about 10 nm [7]. The HPHT NDs synthesized from organic matter using this method do not contain metal impurities and have good biocompatibility as a good material for bioimaging probes. For DNDs and HPHT NDs that contain both NV centers, due to the highly nonequilibrium (detonation synthesis) and near-equilibrium (high pressure and high temperature synthesis) synthesis conditions, the NV color center concentration of DNDs is lower than that of HPHT NDs, making the fluorescence performance of DNDs weaker than that of HPHT NDs [8]. Both types of NDs are able to exhibit high single-photon brightness, even at room temperature, making them good bioimaging probe materials for a wide range of applications in the life sciences. DNDs have greater advantages over HPHT NDs in bioimaging applications because of their smaller size, larger specific surface area, and abundant surface groups, and, therefore, their high drug loading efficiency, allowing them to be used as nanocarriers to deliver various functional molecules such as contrast agents, proteins, nucleic acids, and small molecule drugs [9].

Conventional fluorescent dyes have the disadvantages of photobleaching, interference from the autofluorescence of biological tissues, and low efficiency in infecting cells [10]. Later in the development of the bioimaging field, quantum dots and metal nanoparticles were used as an alternative to traditional fluorescent dyes. However, the fluorescence emitted by such fluorescent agents is relatively weak and accompanied by blinking, which poses a problem for the detection of fluorescence, and their inherent biotoxicity is not conducive to long-term fluorescence detection in living organisms [11]. Nanodiamond is chemically inert, fluorescent but non-photobleaching, and non-toxic, making it a candidate for bioimaging. Bioimaging facilitates the study of intracellular nanoparticle transport pathways, and the entry of NDs into cells is related to the size [12,13] and shape [14] of the NDs, the time that the NDs are co-cultured with the cells [12], and the material with which the NDs are complexed [14]. In many ND-X therapeutic systems (ND-based cancer treatment systems), NDs generally remain in the cell after transporting drug molecules and do not have a significant effect on the cell for a short period of time, but further studies are needed for a prolonged treatment process [15]. Bioimaging provides powerful tools for research in the fields of bio-diagnostics, bio-therapeutics, and drug delivery. This review will introduce several common approaches to bioimaging, summarize the advantages and disadvantages of different approaches, propose more efficient and implementable imaging solutions, and predict the future development of the field of nanodiamond bioimaging, which will hopefully be helpful for future research work.

## 2. Fluorescence Imaging

Fluorescence imaging with NDs covers several aspects. As mentioned above, there are many functional groups on the surface of DNDs and these functional groups can have a significant effect on the properties of DNDs. In recent studies the aggregation stability of DNDs hydrosols has been related to the interaction of charged particles in the functional groups on the surface [16,17]. Dispersion of DND aggregates using a range of modalities contributes to the formation of stable aqueous suspensions of DNDs. For example, DND aggregates can be purified into small particles by powerful sonication and oxidative grinding in water [18]. Annealing the ND aggregate powder (>100 nm) in hydrogen, the aggregates are broken down into their core (4 nm) particles, which are then dispersed into water by high power sonication and high-speed centrifugation, producing monodisperse ND colloids [19]. Further, sp^3^-sp^2^ rehybridization of carbon atoms on the surface of DNDs yielded 4–5 nm individual ND particle aqueous solutions [20], rupturing DNDs agglomerates by deep purification and air annealing followed by centrifugation to produce DND hydrosols with high negative zeta potential [21]. Obtaining stable aqueous suspensions of DNDs of 4–5 nm is of great importance for surface modification of nanoparticles in biomedical applications. Smaller size DNDs can provide better stability and, in one study, small size DNDs (~4 nm) were able to obtain better photonic activity [19]. Surface functionalization of pristine NDs is an important component of fluorescence imaging, and Avdeev et al. reported two types of stabilization of DNDs in aqueous suspensions: negative potential (−stabilization) and positive potential (+stabilization) (achieved by surface modification of DNDs) [16]. Surface functionalization of NDs also involves wrapping a cationic polymer on the NDs surface which can be used to localize and detect the polymeric nanoparticles by using the spectral properties of different polymers. Connecting the NDs with green fluorescent protein (GFP) is also a kind of surface functionalization of NDs, which can track the cells containing the compounds by the fluorescence emitted by luciferase gene expression.s. The optical defect centers contained inside the FNDs (fluorescent nanodiamonds) can also emit fluorescence, and different optical color centers have different characteristics. The study and analysis of photoluminescence spectra can help us to find NDs with suitable size and luminescence intensity, while different optical color centers can be identified by the different ranges of zero-phonon lines appearing on the photoluminescence spectra. Secondly, the fluorescence lifetime of NDs also has a significant impact on bioimaging studies, as fluorescence lifetime imaging provides insight into the distribution of ND particles in biological tissues through lifetime distributions and allows detailed mapping of biological tissue structures [13]. High-resolution imaging tools are essential for biological imaging, and different microscopes have different imaging characteristics. We will introduce different types of imaging microscopes and the latest progress obtained from new microscope imaging techniques.

### 2.1. Photoluminescence Spectroscopy

In general, photoluminescence spectra are obtained by excitation of fluorescent molecules with different wavelengths of laser light [22]. NDs have abundant optical color center defects, with zero-phonon lines at 575 nm and 637 nm for NV centers, 738 nm for SiV centers, and 883 nm and 885 nm for nickel-related centers, respectively [5]. The zero-phonon line of NDs’ optical color centers has a characteristic peak in the photoluminescence spectrum, which allows the analysis of the photoluminescence spectrum to identify what type of vacancy defects are present in the NDs. Irradiation of NDs with He^+^ ions, protons, or high-energy electrons introduces charge vacancies in NDs. The NDs are then annealed at high temperatures (above 700 °C) to move the charge vacancies toward the nitrogen atoms, thereby allowing the nitrogen atoms to replace a carbon atom in the original structure of the NDs, forming NV defects (Figure 1a, inset) [23]. The fact that there is more than one step of heat treatment after annealing, or even immersion in strong oxidizing agents, makes this method very costly, and the FNDs prepared in this way are too large and difficult to be internalized by living cells, meaning further exploration of easy methods for preparing FNDs is needed [24]. In general, this treatment results in two types of NV centers: NV^0^ and NV^−^ (Figure 1a). The zero-phonon lines of the two NV centers are at 575 nm for NV^0^ and at 637 nm for NV^−^, as marked in Figure 1a. Figure 1b shows a PL peak at 738 nm, which is characteristic of the negatively charged SiV center, and spectral fluctuations appearing at 525, 600, 660, and 740 nm are correlated with other optical defect centers of NDs [25]. Both images (Figure 1(c1,c2)) show the presence of a peak at 885 nm, which indicates the vacancy center associated with nickel, also known as the 1.4 eV center, due to the fact that nickel may not be bonded to the adjacent carbon atoms in the NDs, [5]. Looking at the images, we can see that the shape and peak intensity of the PL spectra obtained by excitation of NDs using different wavelengths of laser light are different, but the characteristic spectral lines appear at the same location.

The peak intensity of the PL spectrum is dependent on the size of the NDs, the excitation wavelength of the laser, the temperature, and the conditions of the single- or two-photon excitation utilized. It can be seen in Figure 1b that the photoluminescence intensity of ND particles with an average particle size of 25 nm is lower than the other two types of diamond particles with larger average particle sizes [25]. The effect of temperature on the intensity of PL spectra is shown by the decrease in the fluorescence intensity of NDs with increasing temperature, which is caused by the thermal activation of non-radiative trapping in NDs [26]. In Figure 1(d1), observing the characteristic peak of the 1.4 eV nickel-related center of the spectrum near 883 nm, it can be clearly observed that the larger the ND is, the higher the intensity of the PL peak is [5]. Comparison within Figure 1(d1,d2) reveals that the PL emission peaks at 883–885 nm are essentially the same for the 500 nm and 2.5 µm NDs in both figures. However, the PL spectra of the 2.5 μm ND in Figure 1(d2) show additional peaks appearing in the 845–855 nm range and near 996 nm. These peaks are also due to fluorescence emission from nickel-associated vacancy centers but, unlike the conventional 1.4 eV nickel vacancy centers, these two peaks show PL emission from 1.448–1.466 eV and 1.245 eV nickel vacancy centers, respectively. The study of two-photon excitation can provide a reference for a wider application of nickel vacancy centers. Therefore, the analysis of photoluminescence spectra helps us to find the suitable size of ND particles and the suitable optical color center.

**Figure 1 molecules-28-04063-f001:**
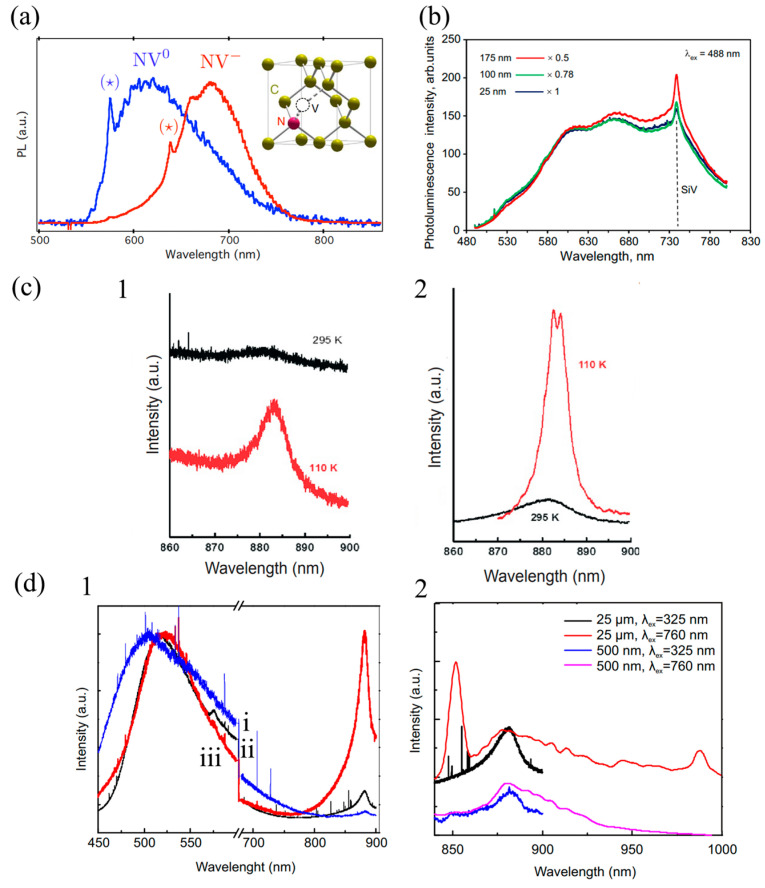
(**a**) Photoluminescence emission spectra of NV^−^ and NV^0^ centers. The star marks the zero-phonon line of NV^−^ (637 nm) and the zero-phonon line of NV^0^ (575 nm). The inset shows the structure of the NV center. [Reprinted with permission from Ref. [27]. Copyright 2010, American Physical Society]. (**b**) PL spectra of Microdiamant™ polycrystalline diamond particles with various submicron fractions: blue—DP 0–0.05 (mean size 25 nm), green—DP 0–0.2 (mean size 100 nm), red—DP 0–0.35 (mean size 175 nm). These spectra were determined under a laser with an excitation wavelength of λ = 488 nm. The prominent peak at 738 nm is the zero-phonon line of negatively charged SiV^−^ centers, which are marked by the vertical dashed line, and it can be observed in all polycrystalline diamond fractions. The spectra are specially normalized for PL intensity at λ = 590 nm in order to produce better comparisons. The figure shows normalizing coefficients. [Reprinted with permission from Ref. [25]. Copyright 2020, Springer US]. (**c**) PL emission spectra of NDs obtained under 532 nm (1) and 325 nm (2) laser excitation with peaks showing fluorescence emission from 1.4 eV nickel-associated vacancy centers (**d**), (1) PL spectra of NDs of different sizes obtained using single-photon excitation with a 325 nm laser at room temperature; (i) 100 nm, (ii) 500 nm, and (iii) 25 μm. (2) PL spectra of NDs of different sizes obtained using two-photon excitation with a 760 nm laser at room temperature, with the sizes of NDs marked on the upper right. [Reprinted with permission from Ref. [5]. Copyright 2018, Nizhny Novgorod State Medical Academy of the Ministry of Health of the Russian Federation].

### 2.2. Imaging Microscope

#### 2.2.1. Optical Microscope

Fluorescence optical microscopy can use the optical color center of NDs to observe and measure the distribution of fluorescent molecules in biological cells or tissues. The fluorescence generated by surface modifications of NDs can also be imaged by optical microscopy. Fluorescence optical microscopy has a wide range of applications in the fields of disease diagnosis, drug delivery studies, and bioimaging. However, optical microscopy is limited by the bounds of light diffraction. In the visible range, the lateral resolution of an ordinary optical microscope is about 200–300 nm and the axial resolution is about 500–700 nm, which means it is not capable of ultra-precision imaging [28]. Next, we will introduce new super-resolution microscopy techniques that have been generated in recent years.

Zhang et al. prepared ND-PDMAEMA brushes by combining inorganic NDs with an organic PDMAEMA (poly-[2-(dimethylamino) ethyl methacrylate]), which can display green fluorescence under 488 nm laser excitation. They used fluorescence microscopy and flow cytometry to determine luciferase gene-infected cells mediated by ND-PDMAEMA brushes and PEI25k, respectively. The comparison revealed that the GFP expression efficiency of ND-PDMAEMA brushes was higher than that of PEI25k [12]. Gold nanoparticles (GNPs) were coupled with NDs to form GNP–nanodiamond pairs and coupled fluorescence emission from NDs could be induced using the harmonic mode emission of gold nanoparticles described by Schmidheini et al. [29]. The researchers studied the optical coupling of GNP-nanodiamond pairs using multiphoton optical microscopy and showed that GNP can efficiently excite NV centers in ND. The study of GNP–nanodiamond pairs contributes to the development of super-resolution microscopy. Multi-photon optical detection has a greater prospect of application in the field of intracellular bioimaging. Liu et al. made FND-gold hybrid particles (FND-Au) by combining FND containing NVs with gold nanoparticles through a controlled protein encapsulation method [30]. A549 cells (an adenocarcinomic human alveolar basal epithelial cell line) were cultured in an FND-Au content of 100 μg/mL for 24 h to obtain confocal laser scanning microscopy (CLSM) images. Imaging the sample in different scattering channels, the resulting images are specific for nanoparticle detection, facilitating the differentiation of nanoparticle signals from organic fluorophores of biological tissues. Analyzing the images, we can get the FND-Au to be dissociated in the cells. The team performed further cultures and experiments to obtain single FND-Au and larger FND-Au coaggregates, which can be used for bioimaging studies.

The principle of confocal fluorescence microscopy is to focus the probe light emitted from a point source onto the observed object through a lens and, if the object is exactly at the focal point, the reflected light converges back to the source through the original lens and is processed to produce an image. Petrakova et al. used confocal microscopy to observe FND-PEI (poly(ethyleneimine))-DNA complexes in IC-21 and HT-29 cells and stained the nuclei with DAPI. The observations showed that the complexes could be distinguished from biological tissue autofluorescence [22]. Perevedentseva et al. used confocal microscopy to obtain confocal fluorescence images which were analyzed to understand the penetration and localization of NDs in cells [13]. Laser scanning confocal microscopy (LSCM) uses a fluorescence microscope retrofitted with a laser scanning device [31]. Zhang et al. used LSCM to investigate the entry of DNA into the nucleus, transported by a complex (ND-PDMAEMA brush ligated with DNA) [12]. LSCM images show that the complex of the ND-PDMAEMA brush attached to DNA has entered the cell; after some time, DNA can enter the nucleus while the ND-PDMAEMA brush stays outside the nucleus. After a longer period of time, about three hours, ND-PDMAEMA brushes were also detected in the nuclei of the cells. This study demonstrates that ND-PDMAEMA brushes transport DNA into infected cells with high efficiency and can enter the nucleus in a short time.

NDs can be used in cancer therapy research to efficiently transport drug molecules to the site of tumor cells and to avoid the breakdown of drug molecules during transport. Perevedentseva et al. summarized the ND-X delivery platform for anticancer drugs. X refers to anti-cancer drugs; the more commonly used anti-cancer drugs include doxorubicin hydrochloride (DOX), cis-dichlorodiammineplatinum (II) (CDDP, cisplatin), etc. [15]. DOX is a chemotherapeutic agent that induces apoptosis in cancer cells. In a study, ND and DOX were first covalently linked and then covered with a pullulan-di-(4,1-hydroxybenzylene) diselenide (Pu-HBSe) shell to produce NDX-CCS, which prevents the aggregation of ND-DOX and the leakage of the drug [15]. In the study, NDX-CCS was injected intravenously into HepG2 tumor-bearing mice, and drug accumulation was found in the tumor cell sections of the mice. The researchers performed confocal laser scanning microscopy on frozen sections of mouse tumors. The images show that the morphology of the drug-treated tumor cells was changed, and their growth was inhibited. In another study, CDDP was attached to the surface of a ND by physical adsorption to create an ND-CDDP complex, which was found to be more effective in acidic cancer cell extracellular fluid or in the cytoplasm of cancer cells [15]. Some studies have also investigated dcHSA-ND complexes made by covering the ND surface with a human serum albumin-based biopolymer (polyethylene glycol) coating. The researchers performed confocal microscopy imaging of the complex as it crossed the blood-brain barrier, neuronal cells, and astrocytes. This image shows that accumulation of dcHSA-NDs occurred in the studied cells [15]. The ND-X transport platform has greater efficiency and stability than drug treatment alone, but NDs may remain in the cells during treatment. While this has negligible effects on cells in the short term, long-term effects have not been proven and further studies are needed.

Wu et al. generated a nanogel (NG) layer on NDs’ surfaces using an adsorption cross-linking method; this nanogel layer was able to provide NDs with more attachment points for bioactive groups and could improve the colloidal stability of NDs in aqueous solution [14]. The PDT agent [4-(1H-imidazo [4,5-f][1,10] phenanthrolin-2-yl-κN7,κN8)benzoato]-bis(2,2′-bipyridyl-κN1,κN1′)ruthenium(1+) chloride (Ru-COOH) is a commonly used anticancer drug that emits singlet oxygen (^1^O_2_) upon excitation by light. ^1^O_2_ is cytotoxic and can inhibit the growth of tumor cells to a large extent. Covalently linking the carboxyl group in Ru-COOH with the amine group in ND-NG makes a ND-NG-Ru complex. HeLa cells were cultured in ND-NG-Ru at 100 μg/mL for 4 h to obtain LSCM images (Figure 2a), which showed that ND-NG-Ru was effectively absorbed by the cells and ND-NGs were mainly distributed in the vesicles of HeLa cells. The cells were stained with fluorescein diacetate/propidium iodide, which showed green color if they were alive and red color if they were dead (Figure 2b). A large degree of apoptosis occurred after ND-NG-Ru treatment. The researchers further verified the apoptosis using Annexin-V (a protein that binds to phosphatidylserine), and the green fluorescent signal in Figure 2c represents apoptotic cells. Thus, it can be demonstrated that ND-NG-Ru treatment can kill cancer cells rapidly and effectively. In recent studies, DNDs were used as carriers of PDT photosensitizers to form novel photocatalytic materials that facilitate further ^1^O_2_ production. Kulvelis et al. used the photosensitizer Radachlorin^®^ (sodium salts of chlorin e_6_, chlorin p_6_, and purpurin 5) with polyvinylpyrrolidone (PVP) and DNDs to synthesize a ternary catalytic complex, using UV irradiating the complex to excite the DNDs, which do not emit light and transfer energy to surrounding molecules, thereby catalyzing the production of ^1^O_2_ [32]. Due to the chemical inertness of the DNDs in this new catalytic complex, they are resistant to ^1^O_2_ and can remain stable for a long time. Lebedev et al. developed a new photoactive catalyst synthesized from europium diphthalocyanine molecules dissolved in dimethylformamide and transferred into the aqueous dispersion of DNDs (~4.5 nm in size and positive ζ-potential of ~30–40 mV) to form diphthalocyanine phthalocyanine-ND complexes [33]. This hybrid structure can be used as a catalyst to produce ^1^O_2_ in the surrounding medium (air, water, biological tissues) under light excitation at wavelengths ~600–700 nm [33].

Petrakova et al. used oxygen-terminal HPHT FNDs containing NV centers combined with polyethyleneimine (PEI) to make FND-PEI complexes [22]. The complex can use the surface of FNDs to interact with charged molecules, thereby causing the charge state of NV centers to switch between negative and neutral charges. The switch in electrical properties is accompanied by a change in the fluorescence color of NV centers, so this phenomenon can be used as a nano charge sensor to detect the efficiency of DNA infected cells and the release of intracellular payloads. The DNA and FND-PEI complexes are subjected to a series of treatments to synthesize FND-PEI-DNA complexes. Transmission bright field images and confocal microscopic images were obtained by culturing IC-21 macrophages in a culture medium containing an FND-PEI-DNA complex at 30, 60, and 120 min. After the FND-PEI-DNA complex infects the cells, the DNA is separated from the complex and acts specifically on the cancer cells for therapeutic purposes. To determine whether DNA can be transcribed and translated smoothly in the cytoplasm after separation from the FND-PEI complex, researchers used the FND-PEI complex combined with a green fluorescent protein expression plasmid (pGFP) to make the FND-PEI-pGFP complex. The experiment was set up as a control group for FND-PEI complex and GFP, and IC-21 cells were cultured in cultures containing FND-PEI-pGFP complex, FND-PEI complex, and GFP, respectively, and the translation of GFP gene into green fluorescent protein after transcription was observed after 48 h. Comparison of the images revealed that the FND-PEI-pGFP complex was able to translate the green fluorescent protein to some extent, thus also demonstrating that the FND-PEI-DNA complex can have active DNA translocation into the target cells and translate the corresponding protein for the purpose of disease treatment. This study facilitates the use of FNDs as nanosensors for the detection of cellular microstructures, which can be better applied for cancer diagnosis and treatment.

#### 2.2.2. Electron Microscope

An electron microscope is a device that uses an electron beam focused by an electron lens to form a microscopic image of an object. The resolution of an electron microscope is much higher than that of an optical microscope. Electron microscopes are mainly divided into three categories: transmission electron microscopes (TEM), scanning electron microscopes (SEM), and scanning transmission electron microscopes (STEM). The advantage of electron microscopy is the high imaging resolution, but it is not capable of imaging biological tissue cells specifically and it is not capable of showing the dynamic distribution of fluorescent molecules within cells and tracking them in real time.

We first introduce transmission electron microscopy (TEM). The resolution of electron microscopy is higher than that of optical microscopy because the de Broglie wave wavelength of electrons is several orders of magnitude shorter than that of light waves. The electrons in TEM pass through the sample and later form a projection of the sample, which requires the sample to be very thin. Before using NDs as fluorescent labeling molecules, their shape and size after functionalization should be considered; the use of TEM can help us visualize the morphological structure of NDs. Terada et al. used silicon dioxide coating to enhance the performance of NDs. The silicon dioxide encapsulation process was performed on hydroxylated DNDs in dry acetone (or dry tetrahydrofuran) in a nitrogen-filled environment to obtain individually dispersed silicon dioxide-coated DNDs with a diameter of about 10 nm [34]. The obtained complexes were able to maintain good colloidal stability in aqueous solutions or solutions with a wide pH range, and the optical transparency of the silicon dioxide coatings could be adjusted with less effect on the spectral characteristics and fluorescence emission intensity of the NDs. For HPHT NDs, the irregular shape (Figure 3a) can be changed after covering with silicon dioxide to eventually achieve a nearly spherical structure (Figure 3b). However, the thicker silicon dioxide coating formed by this method leads to larger particle sizes, which can affect the biocompatibility and dynamic behavior of nanoparticles in biological cells or biological tissues. It is possible to form very thin coatings by using functionalized silanes (such as (3-aminopropyl) triethoxysilane (APTES) and 3-(trimethoxysilyl) propyl methacrylate (TMSPMA)), which can preserve the original shape of NDs. Wu et al. also studied the aggregation of pure NDs without surface treatment using TEM (Figure 3c), indicating that exposed NDs are prone to aggregation [14]. Figure 3d clearly shows the ND-NGs have a non-aggregated, single-particle morphology. The experimental results show that the NG coating can prevent the aggregation of small-sized NDs. When the ND-PDMAEMA brush composite was used to transport DNA, the size of the complex had a strong constraining effect on the endocytosis capacity and the efficiency of gene transfer cells [12]. Therefore, researchers have analyzed the dimensions of ND-PDMAEMA-1, ND-PDMAEMA-2, and PDMAEMA100 using TEM. The analysis indicated that the smaller sizes of the complex particles can be effectively absorbed by the cells. Among them, ND-PDMAEMA-2 has the strongest ability to bind plasmid DNA due to its high positive surface charge content. Liu et al. conducted TEM imaging studies using FND-Au hybrid nanoparticles as imaging localization particles in order to study the rupture and synthesis of endosome membranes at the subcellular level [30]. This clear and thorough TEM imaging study will be of great value for future studies of intracellular dynamic release processes at the subcellular level.

High-resolution transmission electron microscopy (HRTEM) is an improvement and enhancement of TEM. Zhang et al. studied ND-raw, ND-PDMAEMA-1, and ND-PDMAEMA-2 using HRTEM and obtained high-resolution images [12]. Image sharpness is related to the thickness of the polymer layer and can distinguish ND-PDMAEMA-2 polymer brushes from ND crystals based on morphological differences between non-crystalline polymers and crystals. Liu et al. obtained the spacing between FND particles and gold nanoparticles in FND-Au hybrids down to about 1 nm with the help of HRTEM [30]. Such a narrow particle spacing facilitates the tuning of the electron spin state and fluorescence state of FND in FND-Au hybrids. By fluorescence spectral profile analysis combined with dynamic light scattering, the researchers obtained FND-Au complex nanoparticles with a diameter of about 48 nm, and the obtained nanoparticles were able to be preserved at a temperature of 4 °C for about six months [30]. Osipov et al. developed a method to induce recrystallization of DNDs in a short period of time using ethanol in a supercritical fluid state and 5 nm DNDs at high temperature and pressure, which caused the NVs and SiVs in the original DNDs to disappear and be replaced by nitrogen-vacancy-nitrogen (NVN) centers capable of emitting green fluorescence and paramagnetic substitutional nitrogen (P1) centers with electron paramagnetic resonance properties; their sample was marked as D19 [25]. This method can avoid larger energy consumption and contributes to the large-scale mass production of FNDs with pure green fluorescence properties. The tracking of individual FND particles at the subcellular level is a challenge in the field of bioimaging. Han et al. combined correlated light and electron microscopy (CLEM) (which we will describe in detail in the next section) with dark-field energy filtered transmission electron microscopy (EFTEM) to achieve the first characterization of endocytosis in individual FNDs and to confirm the existence of a membrane tunnel connecting the endosome with the cellular membrane [11]. EFTEM relies mainly on electron density imaging for FNDs, and the measured signal-to-noise ratio of EFTEM imaging is higher than that of TEM bright-field images. The study found that the FNDs were not wrapped by the membrane after entering the mitochondria; one possible conjecture is that the membrane wrapping the FNDs fused with the mitochondrial membrane during the process of FNDs entering the mitochondria, though further studies are needed to prove the specific detailed process of FNDs entering the mitochondrial membrane.

Scanning electron microscopy (SEM) first uses an electron lens to reduce the electron beam spot, then uses a deflection system to scan the electron beam over the sample and after that using a series of processing methods to display the sample image. The sample thickness requirement of SEM is not strict. Khalid et al. used a co-flow device to combine FNDs containing NV centers of about 45 nm in diameter with silk fibroin spheres (a natural biopolymer) to form a layer of silk fibroin (SF) on the surface of the NDs, making ND-SF spheres [35]. The researchers studied and analyzed the particle diameter of FND SF spheres and the separation and degradation of SF with the help of SEM. Analysis of the SEM images reveals that the pre-ND-SF spheres are aggregated in a spherical state and the diameter of the aggregate is around 190–850 nm. SEM images show that ND-SF spheres are obviously degraded, and SF is separated from the NDs. The diameter of the nanoparticles decreases gradually after the separation of SF, to a diameter similar to that of pure NDs. The SF-wrapped NDs can improve the fluorescence emission performance and the stability of NDs, and the SF in ND-SF spheres will be degraded and eventually decomposed into amino acids, which can be absorbed by organisms. Perevedentseva et al. studied the penetration of NDs of different sizes (100 nm NDs, 50 nm NDs, and 3–10 nm DNDs) on the skin surface without a surface coating treatment with the aid of SEM [13]. NDs with a size of 10–20 nm entered the epidermal cells mainly through passive penetration, NDs with a size of several tens of nm entered the cells through the trans-follicular route, and NDs with a diameter of several hundred nm were shown to enter the skin cells mainly through hair follicles.

Scanning transmission electron microscopy (STEM) accepts transmitted electrons and requires that the sample be a thin film. Liu et al. used STEM dark-field tomography combined with Bragg scattering, relying on the blinking behavior of FNDs, to develop the ability to observe FNDs in cells at high resolution even in the absence of fluorescence emission from NV centers [30]. Morita et al. used helium ions (He^+^) implanted into NDs to make helium ion NDs (HeNDs) (containing NV centers and activated by annealing at 1073 K for 2 h), which were combined with gold nanoparticles by electron beam-induced reduction to make Au-NDs. The average diameter of the resulting complex was less than 20 nm [36]. By using a combination of high-angle annular dark-field STEM (HAADF-STEM) and STEM-energy dispersive X-ray spectroscopy, STEM-EDX images were obtained in which gold nanoparticles (bright spots) can be clearly identified in the images in the presence of other elements. Therefore, this microscopic imaging technique can be utilized as a novel method for background-free subcellular structure detection that is capable of providing nanoscale spatial resolution. The investigators further explored the potential of Au-NDs as bimodal imaging probes under in vitro conditions by culturing HeLa cells in a culture medium containing Au-NDs, then determined the spatial location distribution of Au-NDs detected by the STEM-EDX method in the vesicular structures of HeLa cells (Figure 4). The element-selective images (after staining) of the corresponding parts in Figure 4e are shown in Figure 4f–k, and the individual Au-ND particles can be distinguished by comparing the images.

#### 2.2.3. Atomic Force Microscope

AFM has a high resolution, generally at the nanometer level, and can be used to obtain information on the surface morphological structure and surface roughness of objects. Compared with electron microscopy, firstly, AFM can provide three-dimensional surface images, while electron microscopy can only provide two-dimensional images. Secondly, electron microscopes need to provide high vacuum conditions to image smoothly, while AFM can operate smoothly under atmospheric pressure or even in liquid environments. Thirdly, electron microscopy, especially transmission electron microscopy, requires a thin film sample, while AFM does not have strict requirements for samples and does not require special treatment of the samples. The many advantages of AFM have laid the foundation for its application in biomolecular and in vivo imaging. Wu et al. established a novel surface modification method for NDs, adsorption crosslinking, which combines the advantages of both non-covalent adsorption linkage and covalent bonding methods [14]. The colloidal stability of NDs was improved by wrapping them with homogeneous nanogels (NG). The AFM images of ND-NGs under liquid conditions are shown in Figure 5. AFM has some non-negligible disadvantages, such as the limitation of its size measurements, which means it cannot scan the whole sample and count every particle on it [37]. Researchers have used X-ray diffraction (XRD) to characterize the overall structure of samples, and we will cover X-ray imaging specifically in a later section. The establishment of a unified database containing the basic parameters of the ND’s core (size, shape, origin, preparation, surface charge) as well as the coating configuration, the biological model employed, and the complex output can help to select the appropriate NDs as needed. Hydrodynamic diameters measured by dynamic light scattering are used to characterize the coated NDs, influenced by the weak interactions between the NDs. AFM can be used as an alternative method to evaluate the dimensions [38].

### 2.3. Fluorescence Lifetime Imaging

Fluorescence lifetime imaging can be used to specifically image biological tissues by exploiting the optical color centers of nanoparticles. We can analyze the different parameters of fluorescence lifetime imaging (spectral shape, spectral intensity, fluorescence lifetime, polarization) to improve the contrast of the images and thus derive images that are informative. In the process of excitation of fluorescent molecules with laser light, after which the laser light is withdrawn, fluorescence intensity will show exponential decay. The fluorescence lifetime is the time taken for the fluorescence intensity I to decay to 1/e of the initial intensity, I_0_. Fluorescence lifetime is a characteristic parameter inherent to the fluorescent molecule, independent of the absolute luminous intensity. Fluorescence lifetime measurement methods can be divided into two types: the time domain method and the frequency domain method. The time domain method is faster and has higher time resolution, and the fluorescence decay curve obtained by this method can directly reflect the fluorescence lifetime of the sample. The frequency domain method requires simpler instrumentation, which is easy to operate and more advantageous when the fluorescence lifetime of the sample is emitted for a longer period of time, and is more widely used at low imaging speeds [39]. Fluorescence lifetime imaging can even measure the complex autofluorescence of biological tissues [40].

Fluorescence lifetime imaging techniques often utilizes NDs, gold nanoparticles, or a composite of the two as fluorescent imaging molecules. Morita et al. used the interaction of composite materials with gold nanoparticles to plasmonically modulate the fluorescence lifetime of NV centers to enable the monitoring of biological tissues [36]. Using FLIM, the differences in the τ_f_ and I_f_ distributions of endocytosed Au-NDs were clearly shown in <τ_f_> maps. It is shown that intracellular Au-NDs with NV centers have a shortening effect on <τ_f_>, and this property can distinguish Au-NDs from HeNDs in FLIM plots based on <τ_f_> values; Au-NPs can attach to HeNDs with different τ_f_ distributions, thus providing the possibility of τ_f_-based dual-mode multicolor imaging. Perevedentseva et al. utilized the very short fluorescence lifetime of 100 nm NDs (less than 1 ns) to distinguish them from the autofluorescence and exogenous fluorescence of biological samples [13]. They used confocal microscopy and FLIM for imaging, and the comparison of the images obtained showed that the distribution of 100 nm ND in the hair follicle structure indicates that ND can penetrate the hair follicle and be localized in the hair follicle structure. Research on using 100 nm NDs to treat hair follicle structures provides a pathway for NDs to penetrate the skin stratum corneum, and the penetration depth of the NDs and their distribution inside the biological structure during the penetration process are influenced by the size of the NDs. Lin et al. prepared hybrid core-shell ND-gold nanoparticles (ND@Au), which are made by using ND cores containing SiV connected to Au shells [41]. In Figure 6a a significant signal was detected, which mainly came from the gold shell, illustrating the advantage of ND@Au as a fluorescent imaging molecule. The strong fluorescence lifetime signal emitted by ND@Au can be detected in the experimental group (with ND@Au injection) by comparing it with the control group (without ND@Au injection) images; this is due to the short fluorescence lifetime of gold nanoparticles and the easy differentiation of the tissue autofluorescence signal. A noteworthy point is that ND@Au injection into biological tissue cells may cause some potential side-effects; the microscopic image of zebrafish larvae in Figure 6d shows some morphological distortions after ND@Au injection. Comparison of the fluorescence lifetime curves in different cases (Figure 6g) demonstrates that the presence of ND@Au can be detected in zebrafish larvae after injection, and the spectra can be used for quantitative analysis of ND@Au.

Fluorescence lifetime imaging involves fluorescence time-gated imaging, which is a time-resolved imaging technique that uses the variability of fluorescence lifetimes of fluorescent molecules to obtain high-contrast images [42]. Time-gated imaging often utilizes FNDs containing NV centers as fluorescent molecules because the fluorescence lifetime of FNDs is different from biological tissues or cells. Time-gated imaging can be used for the ultrastructural detection of biological tissues and the detection of intracellular dynamics in real time. Hui et al. conducted in vitro experiments and simulated in vivo imaging experiments using FNDs containing high densities of NV centers as imaging molecules, and the results obtained from this experiment provide a feasible method for the real-time detection of transplanted stem cells in vivo [43]. The imaging resolution of the time-gated figure is higher because the fluorescence lifetime of Hb in human blood during the experiment is 0.2 ns, which is much shorter than the fluorescence lifetime of FNDs containing NV^−^ (20 ns). The ICCD shutter time was set to 10 ns to efficiently remove the effect of the Hb fluorescence signal, thus increasing the imaging resolution by a factor of 20. Su et al. achieved the first bioimaging and cell tracking of individual human mesenchymal stem cells in non-rodent animal models using fluorescence lifetime imaging microscopy and time-gated fluorescence imaging [44]. They used human serum albumin (HSA) to wrap 100-nm FNDs containing NV^−^ centers through physical adsorption to make HSA-FND, then used HSA-FND to label human placenta choriodecidual membrane-derived mesenchymal stem cells (pcMSCs), which were then injected into a pig through its left internal jugular vein, thus enabling in vivo imaging and quantitative cell tracking. As shown in Figure 7, it can be observed that the fluorescence images with time gating have higher resolution, which provides a great convenience for the localization and tracking of pcMSCs in porcine lung tissue sections. Figure 4 (Figure 7c) shows images of lung tissue sections after repeated laser irradiation, from which it can be observed that, compared with the previous types of organic fluorescent dyes, FNDs have extremely high stability and do not cause photobleaching. The detection of HSA-FND-labeled pcMSCs in porcine tissues is of great value for in vitro histological detection of cells, cell biodistribution, pharmacokinetics, and replacement of conventional fluorescent stains.

### 2.4. Super-Resolution Optical Imaging

The super-resolution optical microscopy technique is able to obtain cellular sub-microscopic structures at electron microscopy resolutions and to specifically image living cells and capture the dynamic changes inside biological cells. Super-resolution optical microscopy techniques include stimulated emission depletion (STED) microscopy and ground-state depletion microscopy (GSD). STED uses the mutual combination of two strictly co-axial excitation and depletion beams for imaging. The higher the intensity of the depletion beam provided to the STED microscope, the higher the resolution that can be achieved [45]. This imaging approach is similar to that of ground-state depletion microscopy (GSD) in that both rely primarily on nonlinear fluorescence responses to improve the resolution of the imaging during actual fluorescence imaging. Another technique, single molecule localization microscopy (SMLMs), mainly exploits the optical blinking or optical switching properties of the imaged fluorescent molecules to improve the resolution for single molecules through the interconversion of temporal and spatial resolutions. SMLMs mainly include photoactivated localization microscopy (PALM) and stochastic optical reconstruction microscopy (STORM). The imaging principle of STORM is similar to that of photoactivated localization microscopy, but with a different scheme for achieving fluorescence emission sparsity within densely labeled samples. Pfender et al. introduced a method based on the spin of NV centers in bulk NDs that facilitates the use of STORM to combine subdiffraction limit imaging with the simultaneous sensing of various physical quantities [46]. Another technique is structural illumination microscopy (SIM), a microscopy technique that relies primarily on generating a modulated excitation pattern in the sample and achieving super-resolution by encoding high-frequency spatial details in the sample into low-frequency beat patterns that can be computationally processed to reveal sample details at subwavelength scales [23]. The aforementioned super-resolution microscopy techniques can also be classified into two categories, deterministic and stochastic imaging, depending on the nature of the fluorescent molecules utilized for imaging. STED, GSD, and SIM can be categorized as deterministic imaging, which is a microscopy technique that mainly relies on lasers to control whether the fluorophore is in the fluorescent or dark state for imaging. In contrast, the SMLM class of microscopy can be categorized as stochastic imaging, because this class of microscopy mainly relies on the random blinking behavior of fluorophores or quantum-controlled optical switching behavior for practical imaging applications.

Li et al. reported the use of FNDs as fluorescent probes for correlative STED and TEM imaging, which have different imaging environments and require different sample preparation procedures but are able to maintain the imaging quality of FNDs over a long period of time [23]. STED-TEM microscopic imaging mainly utilizes the photostability of FNDs, and the different vacancy defects contained in FNDs can emit different colors of fluorescence, which is conducive to the development of two-color contrast imaging by STED. Castelletto et al. investigated the application of NDs in stimulated emission depletion (STED) microscopy [47], where 25 nm NDs containing seven NV centers have been implemented for STED imaging applications in existing studies. In a further study, FNDs containing high concentrations of NV centers were labeled with BSA and imaged using scanning confocal microscopy and STED microscopy, as shown in Figure 8a,b. STED imaging of single BSA-labeled FNDs achieved a resolution of 50 nm in Figure 8b. Simultaneous two-color dynamic imaging can be achieved by using both NV centers and NVN centers in FNDs as fluorescent molecules for STED imaging. In a more extensive study, it was found that the SiV center contained in FNDs, a vacant center with higher charge stability, has the potential to achieve STED imaging at 20 nm with higher laser excitation energy. In the latest study, SiV centers were introduced in 10 nm FNDs, and while the specific properties of this new material need to be further investigated, it can obtain higher fluorescence intensity under the action of a single emitter.

Torelli et al. utilized FNDs as baseline markers for drift correction in SMLM microscopy [45]. Drift correction is the process of correcting image distortions caused by the physical displacement of the microscope stage due to environmental vibrations, electrical noise, etc. For techniques such as SMLM, the effect of stage drift is extremely pronounced. Drift correction can greatly improve the resolution of imaging. The most important feature of FNDs in this study is the ability to modulate their fluorescence through their spin state, which can provide higher imaging resolution and contrast for super resolution imaging. Due to the size limitation, FNDs are currently not the most ideal material for single-molecule imaging, but future research can further promote the application of FNDs in this field by improving their size and quality and providing higher brightness homogeneities in the production of FNDs.

The harmonic surface plasmon modes of GNPs are unstable and undergo blinking behavior during emission, which can seriously affect the photoluminescence of NDs and can cause NDs to appear to fluctuate randomly between fluorescent and dark states [29]. This fluctuation property can be applied to STORM, and the distribution of NDs can be detected by taking multiple STORM images rapidly. Super-resolution microscopy images of the same site in the time series were obtained using STORM. The NDs with random blinking behavior are marked in the images, clearly showing the presence of light and dark state changes in the NDs. ND-GNP conjugates have great potential for application in the study of more biological models for bioimaging. It cannot be ignored that our current study is not perfect enough to specifically and accurately characterize the fluorescence emission fluctuations of NDs; however, the authors also provide a new idea for the development of this direction, i.e., using multiphoton lasers as light sources to generate random fluorescence emission fluctuations by pulsed laser irradiation, then applying it to the advanced STORM device, which may be able to bring unprecedented breakthroughs [29]. In a more in-depth study, the pulsed magnetic field gradient was used to encode the spin-quantum phase of NDs to achieve quantum control of the optical switching properties of NDs, allowing the photoionization of NV^−^ to NV^0^ for single sparse NVs in bulk NDs to be applied to stochastic optical reconstruction microscopy (STORM) studies [47]. A new breakthrough was also achieved in the SMLM study, where 70 nm FNDs containing a high content of NVs (a single FND contains 1000 NV centers) were biotinylated and then combined with magnetic iron oxide nanoparticles to label the cells. After that, SMLM images with a resolution of 17 nm were obtained under laser excitation at 561 nm. This imaging method mainly exploits the collective blinking property of NVs in FNDs at high concentrations and introduces this property into super-resolution imaging by modulating their spin states within the cellular environment for the first time. This method has the drawback of long image acquisition time while obtaining extremely high imaging resolution, and further improvement is needed. If NVs with high brightness in very small size NDs (less than 35 nm) can be achieved, the method will be further applied in the imaging of finer biological structures.

Structural illumination microscopy (SIM) is a wide-field imaging technique with a lateral resolution of 100 nm and an axial resolution of 250 nm. SIM is capable of providing resolutions above the light diffraction limit by a factor of two and can therefore be used for the study of cellular sub-microstructures [48]. The principle of SIM, in brief, is to use a fine fringe pattern as a spatial carrier on a sample and convert the high-frequency components of the sample into the bandwidth of the optical system, thus reconstructing a high-resolution image [49]. Melnikov et al. constructed a 3D moving fringe-structured illumination using a 3D periodic illumination pattern (interfering with three mutually coherent illumination beams), greatly improving the resolution of the imaging [49]. Opstad et al. used 3D SIM to perform super-resolution imaging of the submicroscopic structure of mitochondria labeled with three probes based on the ability of SIM to be compatible with fluorescent probes [48]. This imaging method enables real-time dynamic imaging of mitochondrial sub-microstructures and allows the study of interactions between different zones within the mitochondria. The study of mitochondrial sub-microstructures and dynamic parameters is extremely important for the study of mechanisms of treatment for many human genetic diseases.

### 2.5. Correlative Light and Electron Microscope

Correlative light and electron microscope (CLEM) combines the high resolution of electron microscopy with the specific fluorescence labeling of optical microscopy. CLEM first uses electron microscopy and optical microscopy to image separately during the imaging process, then superimposes the images made by both to form a correlated photoelectron microscopy image. The key to correlating electron microscopy images with optical microscopy lies in the availability of suitable fluorescent target molecules [50]. FNDs used as fluorescent target molecules can adapt to the extreme conditions of EM and LM imaging and can maintain good fluorescence stability. Compared with previous imaging microscopes, CLEM can study the internal structure of biological tissues in greater detail, which is of great help in drug transport and disease diagnosis and treatment in living organisms. Torelli et al. injected FNDs into tumor cells of mice through intravenous injection and performed CLEM imaging assays on them [45]. In the experiment, confocal microscopy images and electron microscopy images were superimposed to obtain CLEM images, in which we were able to obtain both high resolution and targeted fluorescent markers in the cells. Using CLEM imaging, it is possible to characterize the location and distribution of FNDs in subcellular structures such as mitochondria.

Han et al. proposed another method for characterizing untreated FNDs at the single-particle level that combines high-resolution CLEM and dark-field EFTEM techniques [11]. High-resolution CLEM can achieve the goal of combining the advantages of both by combining HRTEM with LM. The researchers studied individual FNDs in HeLa cells using high-resolution CLEM, obtaining images (Figure 9). Since FNDs are optically stable, the resulting images can clearly represent the distribution of FNDs in the cytoplasm and mitochondria. The analysis of the location and number of FNDs in mitochondria is very valuable for research, which is beneficial for exploring new ways of cancer treatment. Co-localization analysis of FNDs by CLEM combined with other high-contrast imaging modalities can help to study the intracellular transport processes and interaction pathways of FNDs, opening the way for the development of efficient drug transport carriers and bioimaging probes in the future. The team also performed tomography scans of the section shown in Figure 9f with high resolution and found ruptures in the mitochondrial membrane located in the upper left and lower right corners of the mitochondria in the image. The method yields HRTEM tomography 3D images, which can be analyzed to achieve a count of the number of FNDs. The images obtained using CLEM in this study achieved unprecedented resolution, leading in the detection and quantification of single-particle FNDs, and this is of great significance for future studies of nanoscale drug transport across membranes and its application in cancer therapy. The study of the potential cytotoxicity of nanoparticles is also at a leading level, providing great help toward revealing the specific effects in the future.

## 3. Raman Imaging

A sp^3^ characteristic Raman peak generally exists at 1332 cm^−1^ in the Raman spectra of NDs. The peak shape is high and narrow, meaning it is not easily affected by surface functional groups and connecting molecules and can be used as a detection method for the distribution of NDs in biological tissues. The researchers performed Raman spectroscopy on pristine, unirradiated, and nitrogen ion-injected samples that had been excited by a 325 nm laser [37]. The analysis of the pristine ND lines reveals a broad peak between 1500 and 1700 cm^−1^, generally referred to as the “G-band”, which is mainly attributed to graphitic carbon, mixed sp^2^ and sp^3^ in-plane vibrations, and split gap defects within the diamond core. The treated-ND (t-ND) after the annealing and oxidation steps appears with a smaller spectral band (G-band) at 1580 cm^−1^, indicating that the oxidation process eliminated the graphite layer [51]. HPHT NDs are generally above 100 nm in size, and their characteristic Raman peaks are very clear and are often used as imaging probes for biological tissue structures [52].

Raman techniques are also often used in studies of cell structure, constituents, and function and can also be used to detect the composition and structure of unknown fluorescent molecules. Lin et al. proposed to use the Raman spectral information of NDs and gold hybrid nanoparticles (ND@Au) combined with Intracellular ND@Au imaging to obtain information such as cellular composition, structure, and location [41]. Xu et al. combined Raman spectroscopy and X-ray photoelectron spectroscopy (XPS) to derive the composition of tND-TZNGB [t-ND doped tellurite germanate glass [51]. The typical Raman peaks of the cells were found to be generally located at 2800–3200 cm^−1^, which is far from the characteristic Raman peaks of NDs, so the Raman detection method can well avoid the influence of background biological tissues [52]. Therefore, Liu et al. used the characteristic Raman peak of NDs to study the interaction between lysozymes and bacteria [53], which is beneficial for the study of antibacterial drugs. Perevedentseva’s team found that NDs can exist in the liver and lungs of mice for a long time through Raman analysis and HRTEM imaging of mouse digestive organ solutions [15]. This provides guidance for future studies.

Raman imaging combines confocal microscopy techniques and laser Raman spectroscopy, where many spectral data exist in the Raman spectral image that can be analyzed chemically and can correspond to each pixel in the chemical image. By integrating the information from both images, a pseudo-color image can be used to characterize the chemical structure and composition of the sample. The Raman image can be in the form of a 1D profile, a 2D image, or a 3D coloring volume. 3D Raman imaging can provide comprehensive and detailed information on the distribution of cellular uptake of NDs [52]. Figure 10 represents the histogram obtained from Raman imaging and fluorescence imaging of A549 cells cultured in ND@Au at a concentration of 20 µg/mL [41]. The Raman signal in Figure 10(Ia) is mainly derived from the stretching vibrations of C-H bonds in lipids, proproteins, and other organic matter in the cell tissue. Figure 10(Ib) reflects the distribution of NDs in ND@Au. The obvious sp^3^ characteristic peak in Figure 10(Ie) (2) indicates the presence of ND@Au. Analyzing the spectral information, it is possible to study the structure and composition as well as the unique functions of the cells, and even to perform a specific analysis of each pixel point in the image. Figure 10(IIb) shows that the fluorescence intensity is constrained by the SiV concentration in the NDs as well as the relative position distribution. Analyzing the overlapping image (Figure 10(IIc)) enables us to obtain information about the position distribution of ND@Au. Further studies observed that the distribution range of ND@Au increases with its concentration. By staining the cytoplasm and nucleus in different colors and then observing where ND@Au (green) is located, we can observe that ND@Au is located in the cytoplasm. Raman imaging plays an important role in many areas of life science research such as cell metabolism and histopathological analysis (e.g., tumor tissue identification).

Zhang et al. combined Raman imaging and photodynamic therapy (PDT) to both clearly characterize the structural information of tissues within cancer cells and detect the treatment process of photodynamic therapy in real time [9]. In Zhang et al.’s study, the photosensitizer chlorin e6 (Ce6) was used as a therapeutic factor for PDT, and Ce6 was covalently linked to NDs by amidation reaction to make NDs-Ce6 complexes for late-stage Raman imaging studies. NDs-Ce6 combines the spectral characteristic of NDs with the therapeutic ability of Ce6 for cancer therapy and becomes a Raman imaging probe with great potential for application. The researchers examined the ROS generation efficiency of NDs-Ce6 using a 660 nm laser. Later, Raman spectroscopy detection of NDs-Ce6 was performed, and the results showed that no new Raman peaks appeared except for the characteristic Raman peaks of NDs, which indicated that the Raman detection of NDs-Ce6 was negligibly influenced by Ce6. The specific location of NDs-Ce6 in HeLa cells can be easily obtained by comparing the two-color Raman images. In the corresponding Raman spectra, the spectrum in the range of 2800–3200 cm^−1^ is due to the symmetric stretching vibration of methyl and methylene groups, while the characteristic Raman peak at 1332 cm^−1^ is from the stretching vibration of the covalent bonds inside the NDs in NDs-Ce6. The above study process can provide a new method for constructing nanoscale therapeutic platforms and shows the great potential of NDs-Ce6 as a Raman probe in the field of bioimaging.

## 4. X-ray Imaging

X-ray diffraction (XRD) is often used to analyze the structure of crystals. Analysis of NDs using XRD can be used to determine if the graphite layer on the surface of NDs is removed completely. In the process of ND preparation and functionalization, impurities such as metal ions are often introduced, and investigation of whether the obtained NDs samples are pure can be done using XRD. Han et al. performed XRD analysis of the samples, which show three characteristic spectral lines of diamond while not showing the spectral lines associated with graphitic carbon, indicating the high purity of the samples [37]. Sundar et al. calculated the peak intensity of the XRD spectral lines, giving the weight ratio relationship of the different components in the composite [54]. Shery’s team analyzed the shape of the aggregates using SAXS (small angle X-ray scattering), showing the scattered X-ray intensity as a function of the scattering vector (data points), and the analysis showed the presence of elongated structures in the aggregates [55]. XRD analysis can help researchers to find fluorescent molecules suitable for bioimaging and can analyze the structure and components of different complexes.

X-ray photoelectron spectroscopy (XPS) uses monochromatic X-rays as a probe to act on the material under testing. The incident X-ray photons interact with the atoms in the material under testing, causing the atoms to photoionize and thus release electrons. The released electrons have the characteristic information of atoms, and by analyzing the characteristic information of these electrons, various information about the atoms in the material can be obtained, such as the type and content of elements, chemical environment, and chemical valence state. The X-ray photon beam is less destructive to the sample, which is beneficial for the study of biological materials and organic molecules. Xu et al. used XPS to study the chemical state and the chemical environment of the constituent elements in tND-TZNGB glass. The analysis of different dimensions of tND-TZNGB showed that the mixture is a promising nanomaterial for a wide range of applications [51]. Han et al. performed XPS analysis in order to verify the elemental composition, bonding state, and nitrogen concentration of the sample [37]. Only the characteristic spectra of carbon, nitrogen, and oxygen were detected in the spectra, indicating that the metal impurities in the samples had been removed. In contrast to the previous XRD analysis, trace amounts of sp^2^ carbon were present in the XPS analysis, indicating that the graphite was not completely removed from the sample. The combination of XPS analysis and visual darkening of the powder color were able to demonstrate that graphitic carbon content in the sample increased with time passing. XRD and XPS analysis of samples are beneficial for the mass production of small size FNDs and provide good fluorescent probes for bioimaging. Sundar et al. also used XPS analysis to study the surface elemental composition of composites and the chemical environment of cobalt, oxygen, and carbon atoms [54]. XPS detection is usually more delicate than XRD, and, when combined, XRD and XPS analyses can complement each other, facilitate the study of the fine structure of biological tissues, and provide an accurate basis for later bioimaging.

High-resolution X-ray imaging (TXM) has been greatly developed in recent years, with the advantages of strong penetration ability, high spatial resolution, the ability to perform three-dimensional imaging, and low requirements for the detection environment. TXM has an important impact on biological tissue research and structural analysis of material components, and researchers have proposed results for improving the imaging contrast. Gold nanoparticles are generally considered to improve the imaging contrast, and Lin verified this property using high-resolution X-ray image analysis [41]. By comparing Figure 11a with Figure 11b, we can clearly discover the ND@Au detected in A549 cells (marked with color stars). It can be seen that this nanoparticle can also present high-resolution images in TXM detection. The experimental results show that ND@Au is suitable for integration with imaging devices and has great potential for future bio-diagnostic and therapeutic research. In future studies TXM can also be applied to imaging human and mouse cells, using a coherent (synchrotron) X-ray source combined with a high-resolution objective to significantly improve the resolution of imaging; combined with tomography, it can be used to render the subcellular structure of biological tissues.

## 5. Magnetic Modulation Fluorescence Imaging

Magnetically modulated fluorescence (MMF) imaging is a novel bioimaging technique for real-time selective observation of FNDs containing NVs, which can well exclude the interference of background signals from biological tissues. MMF is based on the optically detected magnetic resonance technique (ODMR) (optical detection of the spin state of NV^−^ centered electrons [56]). Igarashi et al. constructed an ODMR microscope that enables MMF imaging of various types of cells [57]. Magnetically modulated fluorescence (MMF) imaging is achieved in two main ways. One approach relies on simple subtraction of images with and without a magnetic field. The other is a phase-sensitive locked detection of FND emission modulation, which allows the use of smaller magnetic fields and faster modulation rates [49]. The reason why FNDs containing NVs can be used as good fluorescent molecules for MMF is that NVs have good magneto-optical properties that facilitate optical magnetic resonance detection.

Optical magnetic resonance detection of FNDs shows that the fluorescence intensity of FNDs decreases at 2.87 GHz, indicating that the spin state of electrons in NVs changes under the effect of magnetic resonance [58]. This study lays the foundation for the detection of intracellular FNDs containing NVs using optically detected magnetic resonance techniques. The team also developed a selective imaging protocol (SIP) method, which is based on the principle that the fluorescence emission intensity of NVs in FNDs under microwave (MW) irradiation is modulated and thus shows fluctuations in the fluorescence intensity curve, while the fluorescence intensity of fluorescent beads without NVs is not affected by MW and thus shows a constant state. The researchers compared the expected fluorescence profile with the actual measured fluorescence profile and proved the feasibility of this method. During SIP image acquisition, the MW is controlled to switch between on and off states and the fluorescence image emitted by NVs only can be obtained by processing and subtracting the images obtained from the two states. The researchers used SIP to detect FNDs in HeLa cells and showed that SIP can well exclude the effect of fluorescent bead signals. SIP imaging can specifically observe individual FNDs in a fluorescence spot according to the difference between the lattice direction and the magnetic field direction of each FND. Researchers examined the signals of FNDs in *Caenorhabditis elegans* using SIP technology. Comparing the normal fluorescence images with SIP images, it can be observed that the interference of the tissue autofluorescence signal is completely excluded in SIP images. The characteristic spectral lines of the experimentally obtained ODMR patterns also confirm that the signal comes from the FNDs. In a further study the SIP technique was applied to nude mice in vivo (Figure 12), and the comparison of Figure 12b,c shows that the SIP technique is well applied in mice. Figure 12d shows the change in fluorescence intensity of the two fluorescent substances. Figure 12e shows the ODMR pattern of FNDs; the detection intensity can remain stable for one hour, indicating that continuous in vivo imaging of mice is feasible for at least one hour. The respiration detection of mice also shows that the injection of FNDs and the SIP imaging process have no significant effect on the survival of mice, and the effect of mouse respiration on the SIP detection is also minimal, demonstrating the great potential of SIP detection as a novel in vivo real-time imaging technique.

Sarkar et al., based on phase-sensitive lock-in detection technology, used software to lock in and amplify the signal intensity of each pixel point over time [59]. Since FNDs are able to produce intensity modulation phenomena corresponding to the magnetic field frequency in the presence of an applied magnetic field, while the background signal does not produce this response, background-free imaging can be achieved at a more accurate level by this locked-in computing technique. The effect of a large number of background fluorophores can be seen in Figure 13a. A plot (Figure 13b) of fluorescence intensity versus time for FNDs (red arrow in Figure 13a) in each pixel value shows the modulation effect of the magnetic field on the fluorescence intensity. A plot (Figure 13c) of the fluorescence intensity change curve for background fluorescence (blue arrow in Figure 13a) does not show this response. Note that the peaks at 0.1 Hz and 0.3 Hz in Figure 13d do not appear in Figure 13e, indicating that the technique of computational locking can reduce the effect of background noise. Figure 13f,g eliminate the effect of the phase shift of the reference signal and the modulated signal with the help of the calculation of the square root of the intensity of each pixel using the amplitude of the periodogram of the sine and cosine reference waves at 0.2 Hz. Figure 13h shows the final background-free image obtained by the phase-sensitive lock-in calculation technique, which improves the signal-to-noise ratio of the imaging. To investigate the imaging of the MMF technique in organisms, the team performed MMF imaging with FNDs as the imaging localization factor in the whole mouse anterior lymph nodes. The image was obtained using the computational locking method, which is more sensitive to the motion of the sample than the two-by-two subtraction method and is beneficial for the dynamic detection of the number and distribution of intracellular FNDs.

Jung et al. presented a study that used MMF to track in vivo FNDs nonspecifically encapsulated with *Caenorhabditis elegans* yolk lipoprotein complexes and analyzed their transport process [56]. Su et al. used MMF detection combined with time-gated fluorescence imaging in small porcine organs and tissues for pcMSCs (human placenta choriodecidual membrane-derived mesenchymal stem cells), which were counted and located [44]. The search for human transplanted stem cells in porcine tissues is of great importance for the study of histological assays. MMF can avoid the interference of tissue fluorescence and can detect multiple FNDs at the same time. MMF works well with other imaging methods, thus combining the advantages of different imaging methods. MMF can detect weak fluorescence signals, thus expanding the depth of tissues that can be effectively imaged by FNDs in vivo, and it can also be used to image FNDs of smaller size. MMF offers the possibility for DNDs to be used as fluorescent molecules; because of their smaller size, this class of NDs is more easily dispersed and compatible in vivo, so the study of small size NDs for bioimaging is of great importance.

## 6. Magnetic Resonance Imaging

Electron paramagnetic resonance (EPR) is a magnetic resonance technique originating from the magnetic moments of unpaired electrons and can be used to qualitatively and quantitatively detect the unpaired electrons contained in atoms or molecules and to explore the structural properties of their surroundings. EPR techniques can be combined with fluorescence imaging techniques to detect defects within FNDs and can help explore the synthesis of novel fluorescent imaging particles. Osipov et al. used the EPR technique to detect defects inside NDs before and after sintering [25]. Observing the EPR spectra of sample D19 at different microwave powers, a clear HFS (hyper-fine structure) feature appears in the spectrum, indicating the disappearance of SiV^−^ and NV^−^ color centers of NDs after sintering and the appearance of P1-substituted nitrogen paramagnetic centers.

Nuclear magnetic resonance imaging (NMRI) is a type of magnetic resonance imaging (MRI). NMRI has the advantages of high imaging efficiency and high contrast as well as less damage to biological tissues, and also provides 3D anatomical images; however, the sensitivity of its detection is limited by the conventional coil-based detection method [60] and the weak polarization of nuclear spins at room temperature [42]. In further studies, the ability of the electron spins polarization in the NV^-^ centers has been improved using optical strategies at room temperature. MRI imaging requires a suitable contrast agent to obtain high quality MRI images. Conventional MRI contrast agents (gadolinium (Gd^3+^), manganese (Mn^2+^), iron (Fe^3+^), etc.) improve the visibility of internal structures by shortening the relaxation time of aqueous hydrogen nuclei in biological tissues [61]. DNDs can be used as nanocarriers attached to metal particles to avoid the effects of potential toxicity of metal particles and to enable targeted MRI imaging. There are two main ways of attachment; one is surface modification of DNDs using groups containing the aforementioned particles. For example, the negatively charged fullerenol Gd@C_82_(OH)_X_ (X∼30) was attached with positively charged DNDs to form a complex, which can form a stable chain structure and, due to complexation, has magnetic resonance properties that can improve the imaging contrast of MRI [62]. Lebedev et al. found in neutron scattering experiments that negatively charged Eu diphthalocyanines and DNDs with positive surface potential in aqueous media can be assembled into a complex that is stable in aqueous solution, and DNDs can improve the magnetic and optical activity of Eu diphthalocyanines [63]. Yano et al. used a pre-oxidation step to create abundant hydrophilic carboxyl groups on the surface of NDs, which are able to disperse the particles in aqueous solution. The carboxylated nanodiamond (CND) was then condensed with gadolinium chelate (Gd-DTPA) to obtain Gd-DTPA-CND complexes with a hydrodynamic diameter of about 4–5 nm, which can be used for high-resolution selective imaging of the lymphatic system [64]. Another approach is to graft metal ions (magnetic and luminescent) directly onto the surface of DNDs [63]. Panich et al. successfully prepared suspensions of Gd-DND at different concentrations and their suspensions showed high proton relaxivity. Such suspensions are promising as MRI contrast agents, which can shorten the spin-lattice (T_1_) and spin-spin (T_2_) relaxation times of water protons and increase the signal intensity of T_1_- and T_2_-weighted MRI images [65]. To maintain the stability of the Gd-DND complex in saline and to avoid its further coagulation in blood, the Gd-DND particles were coated with a polyvinylpyrrolidone (PVP) protective shell, and the obtained PVP-Gd-DND contrast agent provided a higher T_1_-weighted hyperintense signal [61]. The high relaxivity and low toxicity of manganese compounds relative to gadolinium (III) ions allow their use as MRI contrast agents. Panich et al. recently prepared high-purity 4–5 nm DND powders, and the reaction of aqueous suspensions of DNDs with aqueous manganese sulfate solutions allowed direct grafting of Mn^2+^ ions onto the surface of DNDs [65,66]. Similarly performing PVP coverage avoids possible coagulation of the particles in the blood. The interaction of ions with the electronic and nuclear spins of NDs accelerates the relaxation of the spin lattice, which facilitates the improvement of MRI imaging contrast [66]. The ability to enhance the signal of ^13^C atoms (≈10 nm in diameter) in FNDs up to 4700 with a nuclear spin-lattice relaxation time of 55 s at a temperature of 1.4 K has been achieved in recent studies, which further facilitates the application of FNDs in MRI [6]. In a study, FNDs with a diameter of 210 nm (after 2 h of hyperpolarization) were examined using the MRI technique (Figure 14) [6]. Figure 14e shows the co-localization image of ^13^C (red-orange) in FNDs (150 mg/mL) injected into the thorax of mice with the ^1^H component (gray) in the thoracic tissue of mice, which clearly shows the location and specific distribution of FNDs. MRI technology has the advantages of having high imaging contrast, being non-invasive, and causing less damage to biological tissues, and is an important part of biological imaging research.

## 7. Cathodoluminescence Imaging

Cathodoluminescence (CL) imaging allows for both the imaging of the structure of biological tissues and the acquisition of fluorescence emitted by the optically defective color centers provided by FNDs. This technique, which uses an accelerated electron beam to detect the fluorescence emitted by a material, is called CL detection. CL imaging was combined with STEM to provide the desired resolution of ultrastructures. STEM-CL is able to achieve sensitivity even for individual nanoparticles, facilitating the identification and localization of specific FNDs during the imaging process. STEM-CL can be used for defective color-centered detection of semiconductor nanoparticles, while the controllable surface functionalization of FNDs can be used to connect molecules that can be specifically recognized, thus introducing STEM-CL technology to the field of bioimaging.

Glenn et al. performed a CL imaging study using three semiconductor NDs [67]. The three NDs provide sufficiently bright and relatively stable luminescence at room temperature. By observing the CL spectra, we can clearly distinguish the different types of luminescence, and the CL spectra are well suited for imaging multiple fluorescent colors at the same time. The group compared different imaging methods (Figure 15). Comparing the three microscopic images, the leftmost SEM image provides high resolution imaging (5 nm) but does not specifically image the fluorescence emitted by the particles. The rightmost confocal fluorescence shows the fluorescence color of different particles, but its imaging is limited by the light diffraction limit and does not provide high resolution imaging. The middle CL image can show both the fluorescence color of the particles and the required imaging resolution, but its imaging capability is mainly limited by the nanoparticle size. In order to investigate the spatial resolution of NP-CL imaging, the team performed SE and CL imaging using LuAG:Ce NPs; judging the comparison of the two images, it can be concluded that CL imaging has high resolution. The larger sizes of NPs appear as a halo phenomenon, due to the scattering of high-energy electrons at a low angle when the electron beam scans the nearby region of the silicon substrate, which is not the case with smaller NPs. It is worth noting that some NPs in the resulting images are small in size (35 nm) but can produce strong CL signals; such NPs can maximize the resolution and contrast of CL imaging and are therefore of great research value. They have counted the size of NPs, since the resolution of CL imaging in this study is mainly affected by the particle size. Analyzing the images, it can be observed that the signal intensity of nanoparticles has only a small perturbation after multiple irradiations, which verifies the stability of semiconductor nanoparticles under multiple irradiations. Therefore, semiconductor nanoparticles can be used as a powerful tool for CL imaging.

Tizei and Kociak obtained hyperspectral CL images formed on NV color centers in NDs using the STEM-CL imaging technique [68]. Hyperspectral CL images require the high thickness of the imaged sample to be less than 200 nm. The acquired spectral and image information can also be used to make full-color representations of ND particles by using the spectral features as a function of position and as a way to observe the spatial variation in CL within biological tissues. Nagarajan et al. used FND particles with a cationic polymer coating as a probe for the internalization pathway of vesicles and STEM-CL analysis and obtained information on the size of the vesicles, as well as the size, number, and type of FNDs contained in the vesicles [58]. The resolution of CL imaging is correlated with the particle size of NDs and the concentration of the contained color-centered defects; therefore, the development of NDs with high color-centered defect concentration and small size is important to improve the efficiency and resolution of CL imaging [67], and the resolution of experiments in techniques combining CL imaging with STEM is limited by the carrier diffusion length [68]. Correlative light and electron microscope imaging is prone to the problem of more difficult co-localization in the same biological tissues due to the use of both optical and electron microscopes for imaging. STEM-CL imaging in a single instrument can avoid this problem.

## 8. Optical Coherence Tomography Imaging

Optical coherence tomography (OCT) is a new laminar imaging technique, a major technological innovation after X-ray coherence tomography (X-CT) and magnetic resonance imaging (MRI), which detects the optical or scattering properties of tissue and gives information on tissue morphology. OCT is a non-invasive imaging technique that can provide information on tissue damage or lesions by examining the morphological structure of subcutaneous tissue. The principle of OCT, in brief, is to compare whether the optical range difference between the reference light and the ballistic and serpentine photons scattered back in the medium is within the coherence length of the light source. If it is within the range, interference can occur, if not, interference cannot occur, so the ballistic and serpentine photons can be extracted, and these two photons will reflect the information of the sample under testing and can be used for imaging studies.

OCT technique can be divided into one-dimensional, two-dimensional, and three-dimensional analysis. One-dimensional analysis can be used to derive reflectance curves, and computational analysis of imaging with biological tissue removed (without the effect of scattering from biological tissue) uses Origin software to roughly localize NDs. Perevedentseva et al. studied the morphological structure of skin samples treated with 100 NDs and DNDs using OCT technique and investigated the accumulation of NDs in the skin tissue and the interaction of NDs with the skin [13]. Researchers obtained two-dimensional images of black skin tissue through OCT technology. The images show a thin layer with large reflectivity, which may be due to adhesion of NDs or accumulation of dead keratinocytes. The 3D images shown in Figure 16 are OCT images of white skin tissue. The use of osmotically active soaking solution can be used to remove excess biological tissue from the skin tissue and reduce the interference to the imaging. Comparing Figure 16a,b, it can be seen that in Figure 16a the high-level scattering area is observed in the skin layer due to the interference factor, and in Figure 16b the high-level scattering is clearly obtained from the skin surface and the hair follicle part (shown by the yellow arrow) after removing the interference signal. The comparison between Figure 16b,c shows a dense signal (blue arrow) starting in the basal layer and ending in the dermis (green arrow) along the hair follicle (Figure 16c), which is analyzed to be due to the 100 ND treatment. This signal is also indicative of the accumulation of NDs in the skin tissue, resulting in a change in the scattering intensity in this region, which increases the imaging contrast in OCT detection. This study demonstrates that NDs can penetrate into skin tissues and provides a preliminary exploration of the interaction between NDs and skin. In future studies, the distribution location of NDs in skin structures can be studied in a controlled manner by surface functionalization of NDs, and the OCT detection method can be combined with the multiple imaging techniques we mentioned above to provide a powerful tool for further nanoparticle safety detection.

## 9. Conclusions and Perspective

In this review, we have summarized NDs optical defective color centers, the classification and synthesis methods of NDs (with a focus on DNDs and HPHT NDs), and the advantages of NDs as bioimaging probes compared to traditional fluorescent stains and quantum dots. We introduce the method of photoluminescence spectroscopy detection for fluorescence imaging. The fluorescence imaging modalities were classified according to different imaging microscopes and presented the advantages and disadvantages of imaging with different microscopes. Fluorescence lifetime imaging and lifetime decay curves are introduced, and fluorescence lifetime imaging is often combined with time-gated imaging techniques to improve the contrast of the imaging. The composition and structure of different fluorescent molecules or biological tissues can be obtained by Raman spectroscopy, XRD, and XPS analysis, and the biodistribution of FNDs and the structure of biological tissues are clearly shown by two- and three-dimensional analysis of Raman mapping. Magnetic modulation fluorescence imaging (MMF) can overcome the autofluorescence of biological tissues through a series of methods. Magnetic resonance imaging (MRI) includes electron paramagnetic resonance spectroscopy (EPR) and nuclear magnetic resonance imaging (NMRI), and we highlight the application of DNDs in MRI contrast agent preparation. Cathodoluminescence (CL) imaging and optical coherence tomography (OCT) are technologies that have been developed relatively rapidly in recent years and have great potential for future development in the field of bioimaging.

Bioimaging using FNDs can be divided, depending on the stage of their study, into the detection of FNDs at the single particle level (mainly the analysis of the submicroscopic structure of cells) and imaging within cells, for example, in HeLa cells or other kinds of cancer cells, in single-celled organisms such as *Caenorhabditis elegans*, in rodents such as mice, or in the organs of small mammals, such as small pig organs. For early cancer diagnosis, bioimaging provides clear imaging of diseased tissues and can provide an intuitive and accurate basis for early cancer detection. Note that the specific effects of nanoparticles such as FNDs on biological tissues when they are present in biological tissues for a long period of time are yet to be further clinically validated. In cancer therapy, FNDs can be used as drug transport carriers. It is needed to emphasize that the above-mentioned imaging methods are not independent of each other, and different methods can be combined to neutralize the advantages of different methods and overcome the shortcomings of individual imaging. The following aspects need to be further explored in the future research development process. At the level of cellular submicroscopic structure, the study of mitochondrial submicroscopic structure is still not fully characterized, and the characterization of mitochondrial submicroscopic structure is beneficial to the etiology of many human genetic diseases. At the single cell level, the study of the transmembrane transport of NDs and the transmembrane transport of drug molecules carried by NDs is of great value for cancer therapy; At the organismal level, the study of the transfer pathway of stem cells is of great importance for the acquisition and culture of stem cells. In short, this review reveals key issues, opportunities, and challenges of the application of NDs in bioimaging. The authors hope that this review may provide some assistance for NDs in bioimaging and inspire further research activities to make more discoveries in the future.

## Figures and Tables

**Figure 2 molecules-28-04063-f002:**
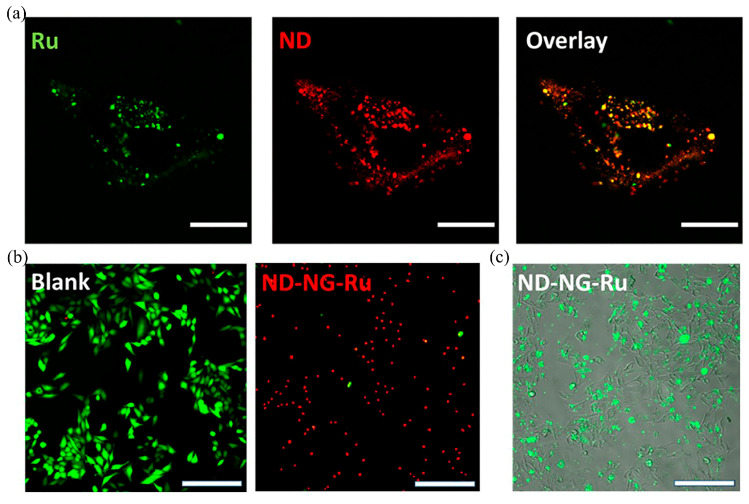
(**a**) HeLa cells were imaged by confocal microscopy after being cultured in ND-NG-Ru at a content of 100 μg mL^−1^ for 4 h (irradiation time: 15 min, scale bar: ¼ 20 μm). Green is the fluorescence signal of Ru and red is the fluorescence signal of the NDs. (**b**) The first panel shows the live cell staining of HeLa cells without ND-NG-Ru treatment (irradiation time: 15 min). The second panel shows the staining of dead cells after treatment with 100 μg mL^−1^ ND-NG-Ru (irradiation time: 15 min). (**c**) Detection of early apoptosis with Annexin V, FITC. Green represents apoptotic cells (scale bar: ¼ 200 μm). [Reprinted with permission from Ref. [14]. Copyright 2021, Wiley-VCH Verlag].

**Figure 3 molecules-28-04063-f003:**
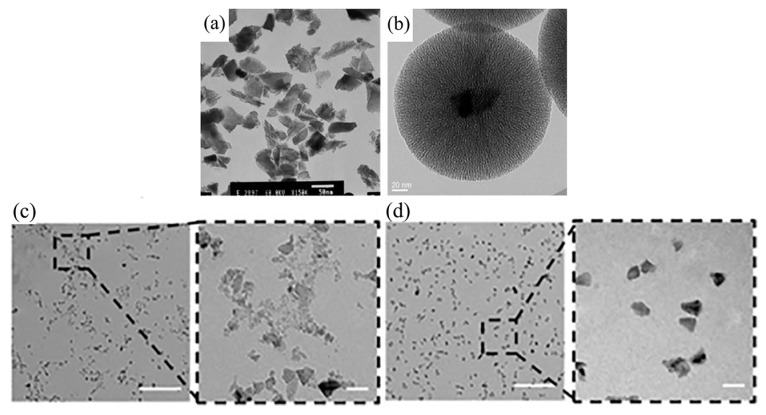
TEM images of silica coated NDs. (**a**) HPHT NDs (**b**) Thick layer of mesoporous silica encapsulating HPHT NDs [Reprinted with permission from Ref. [34]. Copyright 2020, Elsevier]. (**c**,**d**) TEM images of NDs (left: scale bar ¼ 500 nm; right: scale bar ¼ 50 nm) [Reprinted with permission from Ref. [14]. Copyright 2021, Wiley-VCH Verlag].

**Figure 4 molecules-28-04063-f004:**
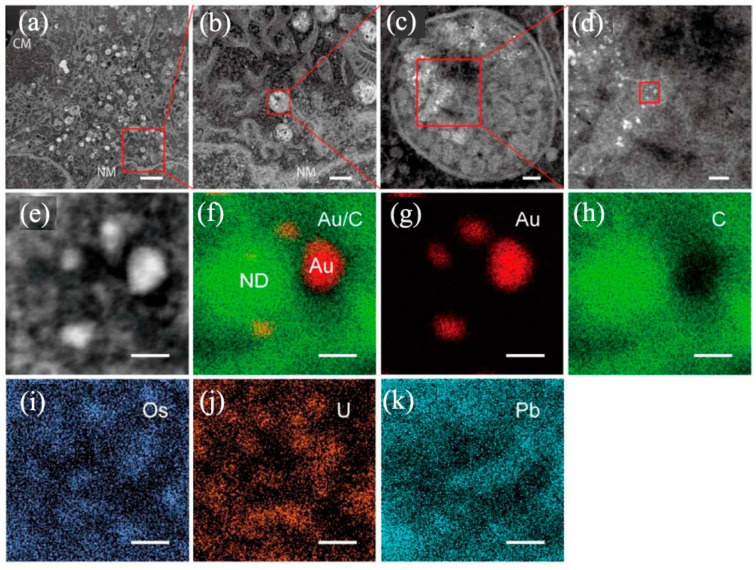
Individual mammalian cells were cultured in medium containing Au-NDs, and the distribution of Au-NDs was detected using microscopic imaging. (**a**) HAADF-STEM images obtained by detecting individual Au-NDs in individual vesicles of HeLa cells at different resolutions. The resolutions of the (**a**–**d**) plots are 2 μm, 500 nm, 100 nm, and 20 nm, respectively. The (**e**) plot is the HAADF-STEM image of the region shown in the red square in (**d**), the (**f**–**k**) plots are the corresponding STEM-EDX elemental maps, and the elements shown in the (**f**–**k**) plots are Au/C, Au, C, Os, U, and Pb. The scale bar of the (**e**–**k**) plots is 5 nm [Reprinted with permission from Ref. [36]. Copyright 2017, American Chemical Society].

**Figure 5 molecules-28-04063-f005:**
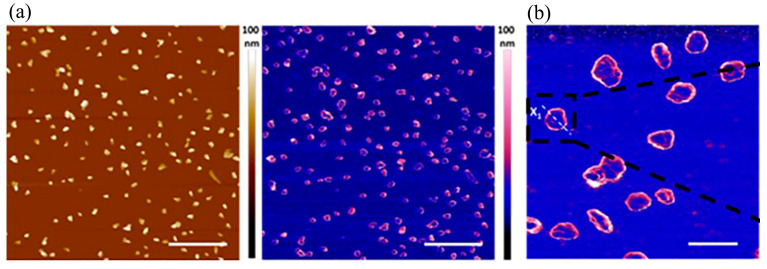
(**a**) AFM images of ND-NGs in liquid state (left: height sensor; right: deformation; scale bar ¼ 500 nm). (**b**) AFM images of ND-NGs (left: scale bar ¼ 100 nm; right: deformation distance profile of ND-NGs) [Reprinted with permission from Ref. [14]. Copyright 2021, Wiley-VCH Verlag].

**Figure 6 molecules-28-04063-f006:**
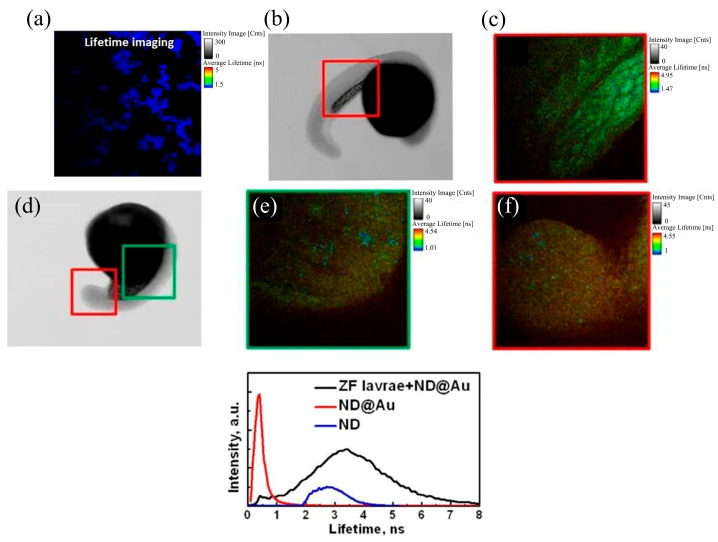
(**a**) Fluorescence lifetime imaging of ND@Au. Two-photon fluorescence of ND@Au is excited by a laser at 800 nm with a laser power ~0.5 mW. Figure (**b**,**c**) are the images of zebrafish larvae without ND@Au treatment, (**b**) is the optical image, (**c**) is the FLIM of the red box part in (**b**). Figures (**d**–**f**) are the images of the experimental group with ND@Au treatment, (**d**) is the optical image, (**e**,**f**) are the FLIM of the green and red boxes in (**d**), respectively. (**g**) histograms of lifetime distribution along the image (**e**) (zebrafish Larvae + ND@Au) and image (**a**) (ND@Au), and for control ND. Fluorescence lifetime imaging of zebrafish larvae was performed under laser excitation at a wavelength of 800 nm with a laser power of 20 mW [Reprinted with permission from Ref. [41]. Copyright 2022, Springer Nature].

**Figure 7 molecules-28-04063-f007:**
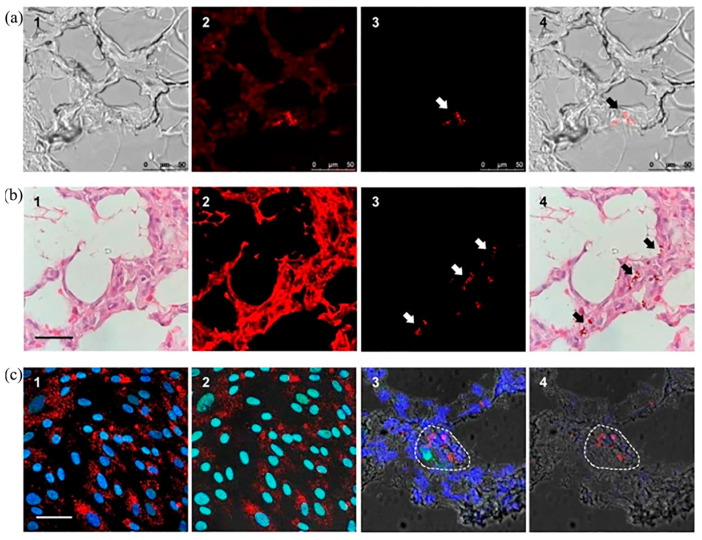
A series of fluorescent images were obtained using HSA-FND labelled pcMSCs in pig tissues. (**a**) shows the screening process of pcMSCs labelled with HSA-FND: (1) bright field optical image, (2) fluorescence image without time gating, (3) confocal fluorescence image obtained after about 8 ns of time gating, (4) overlapping images of (1) and (3). Scale bar: 50 μm. (**b**) Confocal microscopy imaging analysis of pcMSCs containing HSA-FND in lung tissue sections: (1) fluorescence images obtained after staining with H&E stain, (2) fluorescence images without time gating, (3) fluorescence images with time gating, (4) images obtained by overlapping (1) and (3). The white and black arrows in (3) and (4) both indicate pcMSCs labelled by HSA-FND. Scale bar: 20 μm.(**c**) A series of images obtained by immunostaining both pcMSCs in culture medium and pcMSCs in lung tissue sections: (1) Hoechst 33,342 dye stains nuclei blue, FNDs stain nuclei red, (2) cells in (1) stained green again using FITC-tagged antibodies, (3) images of stained lung tissue sections in (2), and (4) only the red fluorescent signal of the FNDs was observed after repeated laser irradiation, while the other organic dyes underwent photobleaching. Scale bar: 20 μm [Reprinted with permission from Ref. [44]. Copyright 2017, Springer Nature].

**Figure 8 molecules-28-04063-f008:**
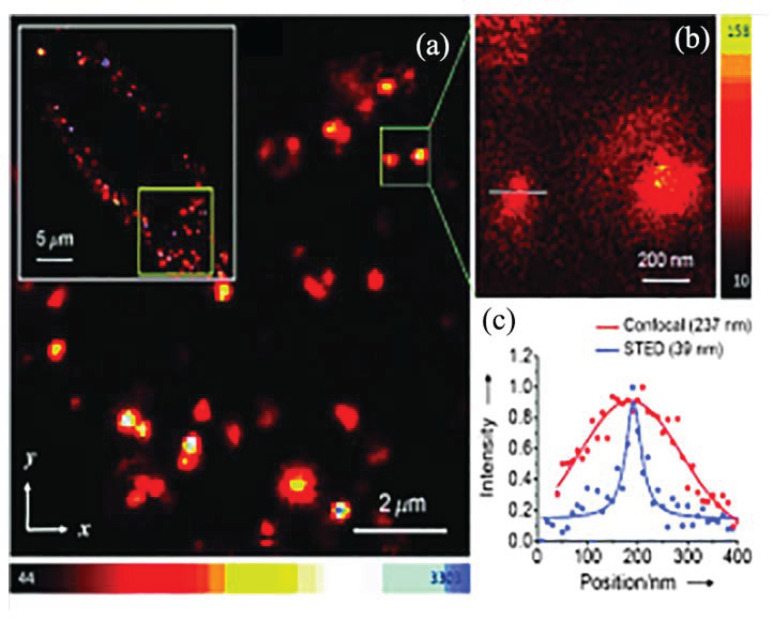
STED images of FNDs and green FNDs in cells. (**a**) White rectangles in the figure show images obtained from scanning confocal imaging assays of BSA bound to NV^−^ FND labeled cells. The green rectangle shows the STED image obtained after binding of a closed individual BSA to an FND. The (**b**) figure shows an enlargement of the green rectangular part of the (**a**) figure. (**c**) The confocal and STED fluorescence intensity curves of the FND shown in (**b**). The solid curve represents the best fit of the Gaussian (confocal) or Lorentzian (STED) function with a FWHM [Reprinted with permission from Ref. [6]. Copyright 2021, Elsevier].

**Figure 9 molecules-28-04063-f009:**
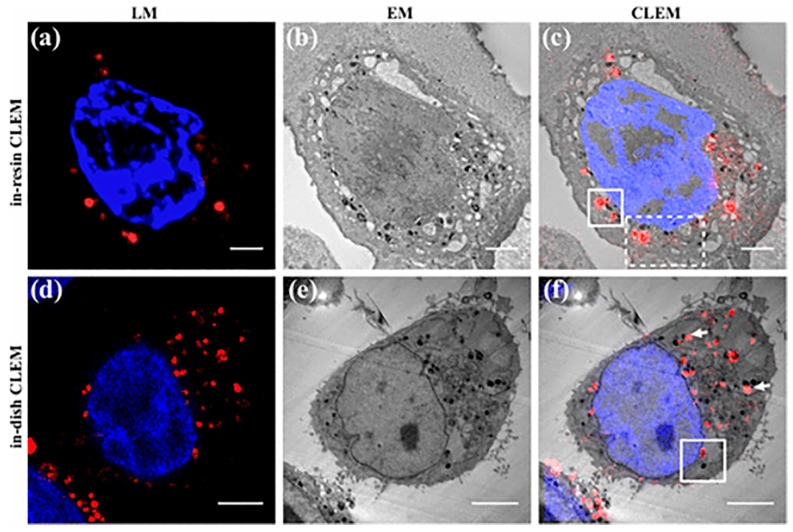
CLEM images of FNDs in HeLa cells inside the resin (**top** image) and inside the disc (**bottom** image). (**a**) CLSM imaging of ultrathin sections of cell samples (~120 nm thickness) with Hoechst stain to stain the nuclei blue and FNDs showing red fluorescent signals. (**b**) TEM image of the same area in (**a**). (**c**) Overlay of (**a**,**b**) two images; the white boxed part refers to the area shown in Figure 3 below. (**d**) HeLa cells were fixed using paraformaldehyde (PFA) and imaged using fluorescence light microscopy, and the nuclei appeared blue after Hoechst staining while FNDs appeared red. (**e**) Electron microscopy images of epoxy sections of the same cells in (**d**). (**f**) CLEM image obtained by superimposing the fluorescence image and electron microscopy image; the white boxed part refers to the area shown in Figure 2 below. The scale bar of the upper image is 2 μm and the scale bar of the lower image is 5 μm [Reprinted with permission from Ref. [11]. Copyright 2019, American Chemical Society].

**Figure 10 molecules-28-04063-f010:**
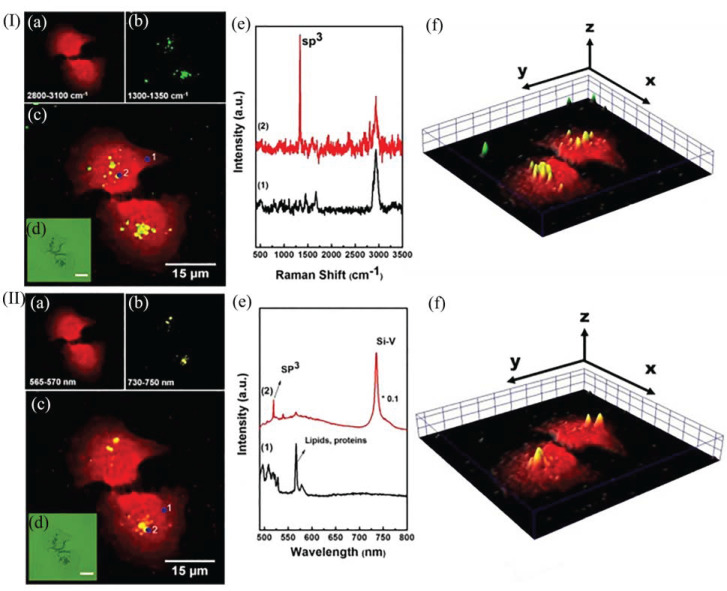
Characterization of the interaction of ND@Au with A549 cells using Raman imaging and fluorescence images. (**I**): (**a**) Intracellular Raman signal intensity distribution maps in the range of 2800–3100 cm^−1^. (**b**) Display plot of ND@Au NP using ND Raman signal in the range of 1300–1350 cm^−1^. (**c**) Overlay of (**a**,**b**) two plots. (**d**) Optical images of the same cells. (**e**) Raman spectra obtained of cell 1 and cell 2 (with ND@Au) in figure (**c**) measured. (**f**) Three-dimensional Raman diagram obtained by reconstruction of the intensity of the obtained spectral signal. (**IIa**) Raman signal intensity distribution within the cells in the range of 565–575 nm. (**b**) Fluorescence characterization of ND@Au NP using the ND SiV center in the range of 730–750 nm. (**c**) Overlay of (**a**,**b**) two plots. (**d**) Optical images of the same cells. (**e**) Raman spectra obtained by measuring for cell 1 and cell 2 (with ND@Au) in figure (**c**). (**f**) Three-dimensional Raman diagram obtained by reconstruction of the intensity of the obtained spectral signal. The excitation wavelength of the laser was 488 nm. The concentration of ND@Au was 20 μg/mL [Reprinted with permission from Ref. [41]. Copyright 2022, Springer Nature].

**Figure 11 molecules-28-04063-f011:**
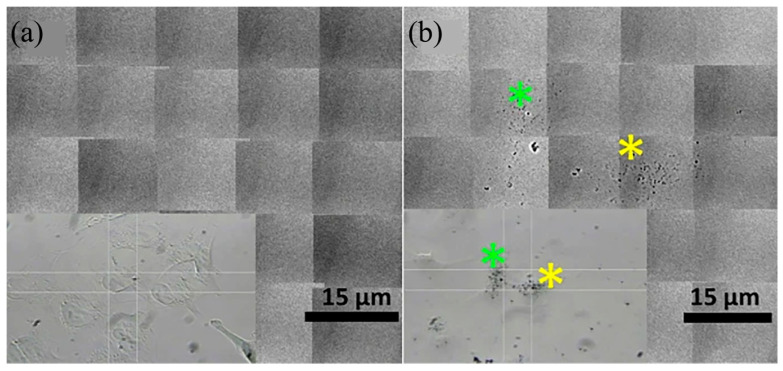
High-resolution X-ray microscopic images obtained by culturing A549 lung cancer cells in medium containing ND@Au. (**a**) Control cell images. (**b**) Cells obtained after culturing A549 lung cancer cells in medium containing ND@Au at 50 µg/mL for 24 h. Colored stars indicate the areas of the X-ray microscopy images corresponding to the optical images [Reprinted with permission from Ref. [41]. Copyright 2022, Springer Nature].

**Figure 12 molecules-28-04063-f012:**
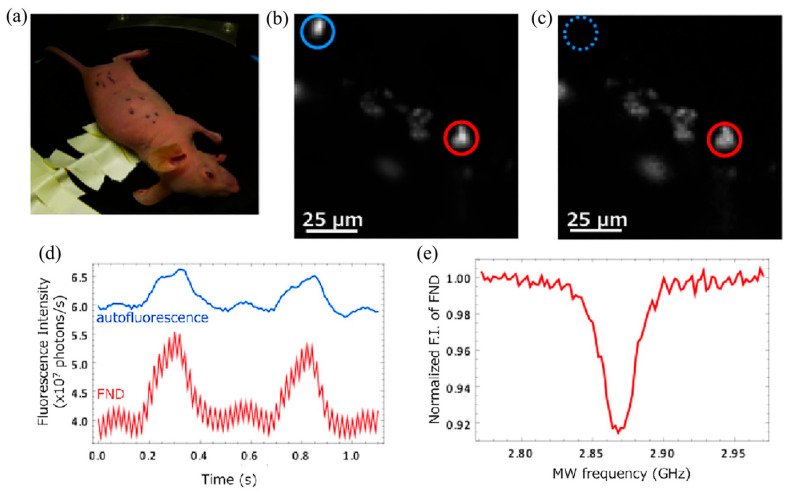
SIP images obtained by injecting ND aggregates into nude mice. (**a**) Photograph of a nude mouse, (**b**,**c**) magnified fluorescence image of a mouse. (**b**) Without SIP processing, (**c**) with SIP processing (EMCCD exposure time of 10 ms). (**c**) The figure shows that the part shown by red circles is the fluorescence signal of FNDs and the blue circles show the fluorescence from other sources. (**d**) The curve of fluorescence intensity with time, autofluorescence curve (blue), FND fluorescence curve (red). Respiration of mice caused the fluorescence signal to show a continuous slow change. The rapid modulation phenomenon of FND aggregates was caused by turning on and off the SIP of MW irradiation every 10 ms. (**e**) ODMR spectra of FND aggregates, corresponding to the red circles shown in Figure (**b**,**c**) [Reprinted with permission from Ref. [57]. Copyright 2012, American Chemical Society].

**Figure 13 molecules-28-04063-f013:**
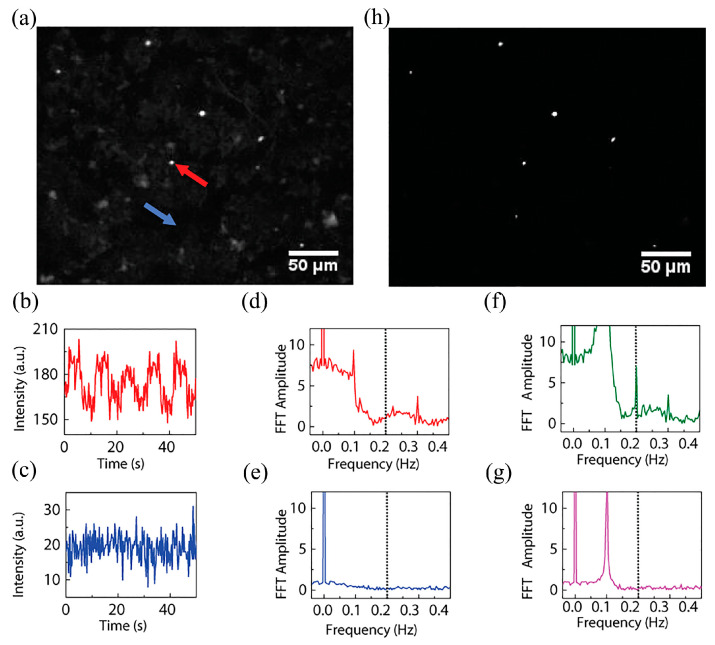
Excluding interference from background signals using phase-sensitive lock-in detection in a wide field of view. (**a**) Scanning confocal microscopy images. A frame from 1000 movies was selected and recorded for 0.25 s per frame (recording conditions: 0.1 Hz modulation of a magnetic field of about 100 G). (**b**) Indicates the curve of fluorescence intensity of the FND indicated by the red arrow in (**a**) as a function of time. (**c**) Curve of fluorescence intensity versus time corresponding to the background signal indicated by the blue arrow in (**a**). (**d**) FFT of the fluorescence intensity versus time for the plot in (**b**), with its magnitude as a function of frequency after transformation. The characteristic peaks appear at 0.1 and 0.3 Hz. (**e**) FFT of the fluorescence intensity versus time in the plot of (**c**), with its magnitude as a function about frequency. (**f**) Multiplying the amplitude value obtained after the FFT in (**d**) plot by the reference sine wave, 1 + sin(2π·0.1·t). A characteristic peak occurs at 0.2 Hz, which corresponds to twice the modulation frequency. (**g**) Multiply the amplitude value obtained after the FFT in (**e**) plot by the reference cosine, 1 + cos(2π·0.1·t). The average of the three points with sine and cosine FFT amplitudes around 0.2 Hz for each pixel is calculated and summed orthogonally, i.e., I = √ (I^2^_sin_ + I^2^_cos_). (**h**) The final image obtained in a wide field of view excluding the background signal. Before the lock-in algorithm processing, the pixel averages of the 1000-frame movie are 173 and 18, corresponding to the pixels of the signals shown in (**b**,**c**), and the corresponding signal-to-background ratio is ~10. After the lock-in algorithm processing, the averages of the FFT amplitudes of the three points near 0.2 Hz are 21.42 and 0.28, corresponding to the pixels of the signals shown in (**b**,**c**) of pixels with corresponding signal-to-background ratios of ~77 [Reprinted with permission from Ref. [59]. Copyright 2014, The Optical Society].

**Figure 14 molecules-28-04063-f014:**
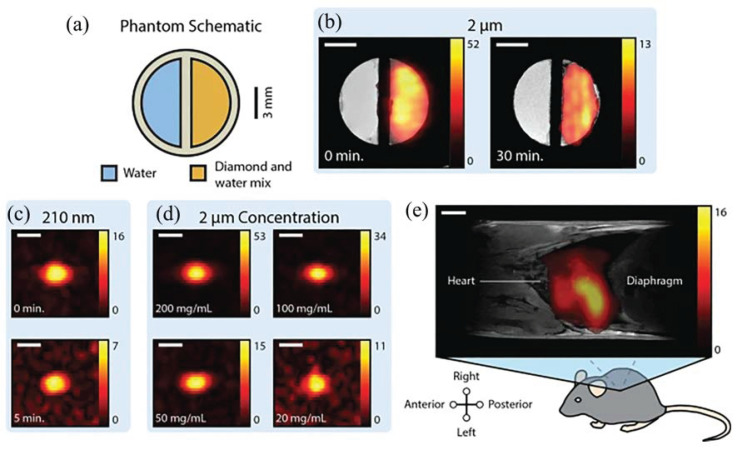
A series of images obtained by injecting hyperpolarized diamonds into mice and then subjecting them to magnetic resonance imaging studies. (**a**) The blue portion on the left side of the “half-moon” model indicates the results of the water-only assay, and the portion on the right side shows orange color indicates the mixture of hyperpolarized diamond and water. (**b**) The “half-moon” model obtained after 0 and 30 min, respectively, of a solution containing 2 μm hyperpolarized diamond content of 120 mg/mL as a sample to the MRI scanner. (**c**) Images of Teflon tubes obtained after 0 and 5 min in an MRI scanner with a solution containing 2 μm hyperpolarized diamonds at 200 mg/mL as a sample. (**d**) A set of images obtained by transferring 2 μm hyperpolarized diamonds from the polarizer into Teflon tubes with different concentrations. (**e**) ^1^H:^13^C NMR images obtained by injecting 2 μm diamonds into the thorax of mice. The gray ^1^H component indicates the location of the water and the structure of the phantom, and the red-orange ^13^C component shows the location of the hyperpolarized diamonds. Scale bar: 3 mm [Reprinted with permission from Ref. [6]. Copyright 2021, Elsevier].

**Figure 15 molecules-28-04063-f015:**
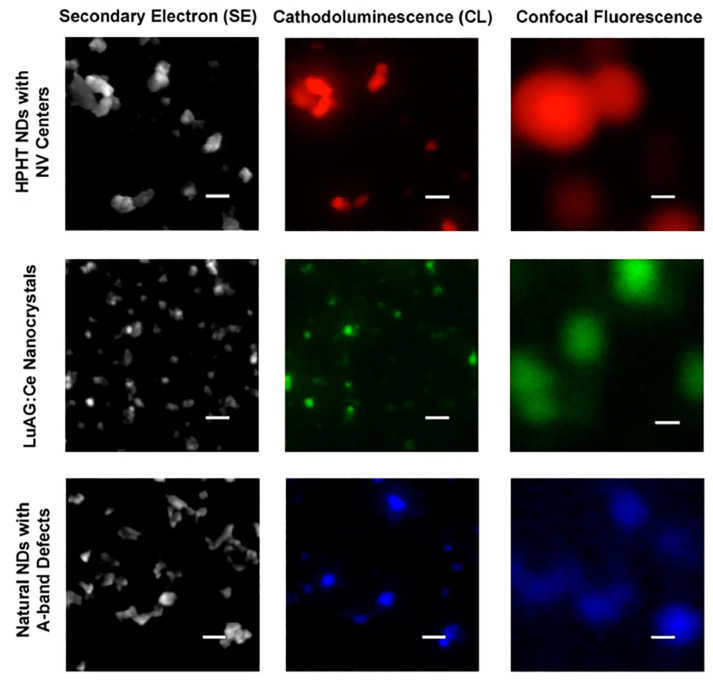
The images are shown from left to right for secondary electron (SE), cathodoluminescence (CL), and confocal fluorescence. The fluorescence emission signals of three semiconductors with red, yellow, and blue colors are shown from top to bottom, respectively. The first column has a high imaging quality with a spatial resolution of 5 nm but does not show the fluorescence color. The middle column is capable of both high imaging resolution and specific fluorescence signals. However, in general the resolution of cathodoluminescence imaging is also limited by the size of the nanoparticles. The third column is able to show fluorescence color but obviously its imaging resolution is not sufficient, and its imaging resolution is limited by diffraction. The scale bar is 200 nm [Reprinted with permission from Ref. [67]. Copyright 2012, Springer Nature].

**Figure 16 molecules-28-04063-f016:**
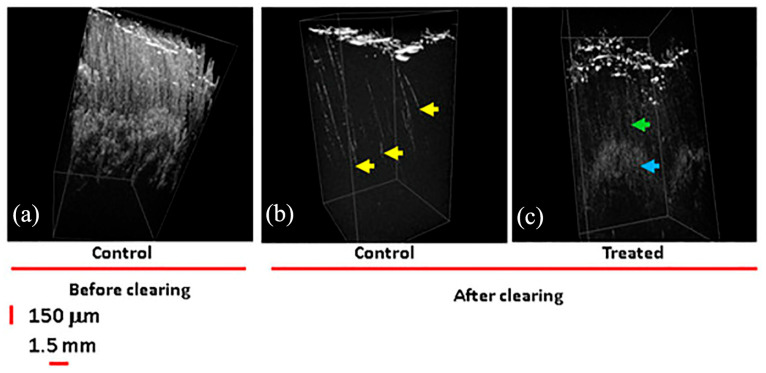
3D OCT images obtained from the detection of white skin samples. (**a**) Control sample treated with PBS before cleanup. Figure (**b**,**c**) are both images after clearing. (**b**) is the control and (**c**) is the sample after 100ND treatment. Both groups of samples show high levels of scattering on the skin layer. The areas of high scattering are distributed in different layers of the epidermis: along the hair shaft (dense, green arrows) and in the hair follicles (cloudy, blue arrows) [Reprinted with permission from Ref. [13]. Copyright 2019, MDPI (Basel, Switzerland)].

## Data Availability

The data presented in this study are contained within the article.

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
