# Peer review of "Multiple Bioimaging Applications Based on the Excellent Properties of Nanodiamond: A Review"

_molecules, 2023, doi:10.3390/molecules28104063_

Round 1
Reviewer 1 Report
Journal: Molecules
Manuscript ID: molecules-2379190
Title: Application of Nanodiamonds in Bioimaging: A review
Wang et al. aim to review the classification of nanodiamonds, synthesis methods, optical defective color centers, nanodiamond surface modifications, and the advantages of nanodiamonds as bioimaging probes compared to traditional fluorescent stains and quantum dots. Several different types of imaging methods were presented in detail. This manuscript has a good idea and a very up-to-date topic. However, it is very long, which makes it extremely difficult to read. The sentences are also very long and confusing. The organization of the text is generally good, but it would be useful to include some critical discussion in the text. In the existing format, the work is more limited to listing the available literature. Also, I think that 50 references are too few for such a long text. It is necessary to shorten and improve the text in order to be published.
The language is suitable.
Author Response
Response to Reviewer 1 Comments
Referee #1:
Comments to the Author
Wang et al. aim to review the classification of nanodiamonds, synthesis methods, optical defective color centers, nanodiamond surface modifications, and the advantages of nanodiamonds as bioimaging probes compared to traditional fluorescent stains and quantum dots. Several different types of imaging methods were presented in detail. This manuscript has a good idea and a very up-to-date topic. However, it is very long, which makes it extremely difficult to read. The sentences are also very long and confusing. The organization of the text is generally good, but it would be useful to include some critical discussion in the text. In the existing format, the work is more limited to listing the available literature. Also, I think that 50 references are too few for such a long text. It is necessary to shorten and improve the text in order to be published.
Authors’ response: Thank you very much for the questions raised by the reviewer. We have checked the entire text and removed some less important sentences and listed the deletions below. I am so sorry that it was difficult to make a large reduction in the length of the article because there are 32 images and explaining each image clearly requires a certain amount of space, and a brief introduction to the different imaging methods and imaging microscopes is necessary. We divided the article into 9 parts, except for the introduction and conclusion part, introduced fluorescence imaging and other 7 imaging methods. In addition to fluorescence imaging, the remaining 6 imaging methods generally contain 2-3 figures, the length of each method is about 3-4 paragraphs. For fluorescence imaging we have made a detailed classification, which for different types of microscopies in the narrative, the reader can find in the reading process, basically each paragraph corresponds to the classification of the preamble, and selective reading according to the order of the article can facilitate the reader's reading.
We have added some critical discussion in the Introduction section on the potential impact of NDs as drug molecular carriers in cancer therapy when present in vivo for long periods of time. In the section 2.1. Photoluminescence spectroscopy, we add the problems in the method of synthesizing NV centers in NDs, and in the section 2.2.3. Atomic force microscope, we add the prospect of establishing a unified database of NDs and the possible application of AFM in this context. We list some of the changes below.
The article provides additional explanations where the elaboration is not detailed enough, and inserts corresponding references, adding a total of 18 references.
Change#1: 1. Introduction
Delete “Both types of NDs are able to exhibit high single-photon brightness, even at room temperature, making them good bioimaging probe materials for a wide range of applications in the life sciences.”
Delete “Detonation nanodiamonds (DNDs) and High temperature high pressure (HPHT) NDs are mixtures of different types of carbon elements and their surfaces are mostly covered by graphite shells, which can be removed by surface purification using strong acidic substances. The surface modification of nanodiamond will help to deepen our understanding of the applications of nanodiamond in the biological field.”
Change#2: 2.2.1. Optical microscope
Delete “LSCM is a powerful tool in the field of biological imaging. The high imaging sensitivity of LSCM allows NDs with natural defects or impurities to be detected by LSCM even if they emit weak fluorescence.”
Delete “and in previous studies DOX was covalently attached to DNDs with diameters of 2-8 nm,”
Delete “However, the study did not compare the therapeutic effect of ND-DOX with the effect of DOC alone.”
Change#3: 3. Raman imaging
Delete “The D, G peak of NDs with small size (<50 nm) broadens and affects the identification of the characteristic Raman peak at 1332 cm-1. In this case, the D, G peak can also be used as a characteristic spectrum to identify the presence of NDs, but this method is not commonly used.”
Change#4: 1. Introduction
In many ND-X therapeutic systems (NDs-based cancer treatment systems), NDs generally remain in the cell after transporting drug molecules and do not have a significant effect on the cell for a short period of time, but further studies are needed for a prolonged treatment process [15].
- Perevedentseva, E., Lin, Y. C., & Cheng, C. L. (2021). A review of recent advances in nanodiamond-mediated drug delivery in cancer. Expert Opinion on Drug Delivery, 18(3), 369-382.
Change#5: 2.1. Photoluminescence spectroscopy
The fact that there is more than one step of heat treatment after annealing, or even immersion in strong oxidizing agents, makes this method very costly, and the FNDs prepared in this way are too large and difficult to be internalized by living cells, and further exploration of easy methods for preparing FNDs is needed [24].
- Chen, J., Liu, M., Huang, Q., Huang, L., Huang, H., Deng, F., ... & Wei, Y. (2018). Facile preparation of fluorescent nanodiamond-based polymer composites through a metal-free photo-initiated RAFT process and their cellular imaging. Chemical Engineering Journal, 337, 82-90.
Change#6: 2.2.3. Atomic force microscope
NDs are heterogeneous materials and modifications of NDs (size, shape, coating, sp2 amount, etc.) have specific effects on carrier behavior and final properties. The establishment of a unified database containing the basic parameters of the NDs core (size, shape, origin, preparation, surface charge) as well as the coating configuration, the biological model employed and the complex output can help to select the appropriate NDs as needed. Using hydrodynamic diameters measured by dynamic light scattering to characterize the coated NDs is influenced by the weak interactions between the NDs. AFM can be used as an alternative method to evaluate the dimensions [41].
- Benson, V., & Amini, A. (2020). Why nanodiamond carriers manage to overcome drug resistance in cancer. Cancer Drug Resistance, 3(4), 854.
Finally, we thank the editor and reviewers very much again for everything done for us. The insightful comments improved the quality of this manuscript. We hope the revised manuscript is acceptable for publication.
Thank you for your consideration.
Yours Sincerely,
Dr. Dandan Sang
Reviewer 2 Report
The manuscript by Xinyue Wang et al. “Application of Nanodiamonds in Bioimaging: A review” presents a review of recent works on nanodiamonds applications in the field of bioimaging. The works on this subject are well documented. However, some questions arise along the reading, and some formulations need a clarification, especially in the introduction section.
1. The introduction requires a brief description of NDs types and way of their producing. DNDs and HPHT NDs are only mentioned (line 58), and need some more insight of their nature – typical particle sizes, concentrations of fluorescent centers, applications, etc. Are their any other types of ND? The classification of NDs declared to be summarized (line 1698) in fact is not presented.
2. The advantages of NDs for bioimaging (lines 74-83) is too brief and need an extended description. At least, two aspects are necessary to introduce. 1) Does the particle size of NDs, larger than many of fluorescent agent molecules, allow NDs to be effectively used as nanocarriers, penetrate through cellular membranes? 2) Are the NDs able to eliminate from the cells and tissues after therapy. To my knowledge, this task is not yet properly solved, and if NDs will accumulate in tissues without excretion, it may cause harmful consequences. This is not the subject of the manuscript, but some introduction into the problem need to be described.
3. A misprint at the line 84: Fluorescence imaging.
4. The description at lines 85-87 is not correct: “For individual detonation 85 NDs due to its small size is prone to aggregation, and the fluorescence emitted from 86 smaller size NDs is weaker and cannot avoid the interference of tissue autofluorescence, 87 which affects the final imaging quality.” At least, can you provide a reference, where DNDs are prone to aggregation due to their small particle size? On the contrary – smaller size (typically ~ 5 nm) provides better stability, which cannot be achieved in the case of large micron and sub microscale particles. This is generally right for many types of particles, not only diamonds. The stable aqueous DND suspensions have been reported repeatedly (ex., doi: 10.1016/j.cplett.2016.06.010, 10.1039/d0cp05533f, 10.1134/S1027451012100151, 10.1021/nn100748k, 10.1016/j.carbon.2017.07.013, 10.1166/nnl.2011.1122). The paper 10.1021/nn100748k even declares that the size for the better photonic activity should be small, ~ 4 nm. DND can agglomerate and form clusters, because of other reasons, not small size – covalent bonds, coulombic and van der Waals interactions (10.1016/j.molliq.2022.118816, 10.1021/acs.jpcc.9b03175), which can be overcome after the process of deagglomeration. DND are prone to aggregation in saline buffers, including isotonic buffer, which can be the barrier for their medical use, but this problem has already been solved (10.1016/j.diamond.2018.05.012). In fact, the main reason for DNDs poor fluorescence is lower concentration of defects, compared to HPHT diamonds. This was reported in numerous works, one of the latest - 10.1016/j.diamond.2023.109754. The links are for your information, I don’t ask to cite them all.
5. “FNDs” are not introduced at the first time it appears, line 93. Are FNDs part of HPHT diamonds, or they are produced by another way? At the line 1482 the authors state that DNDs are the class of FNDs, which leaves some doubts.
6. GNP is not introduced at the first mentioning, line 230.
7. PEI is wrong reported as polyetherimide, line 271. In fact, the authors at the ref. 9 used poly(ethyleneimine).
8. At the line 529 some word is redundant (3D tomographic tomography).
9. At lines 600-602 the authors again describe wrong reason for DND agglomeration. The measured particle size 258 ± 60nm is surely the size of agglomerates. DND very rarely exist in the form of single particles, even in the stable suspensions, where they form chain clusters (10.1016/j.carbon.2016.12.007).
10. A probable misprint, line 997: “GNDs in NDs-GNP conjugates” should be “GNPs in NDs-GNP conjugates”.
11. I may suggest to follow some articles dealing with imaging of DNDs for possible consideration in the review to make it a little more complete, and to demonstrate that DNDs have some certain advances over HPHT diamonds, mainly due to DNDs small size and surface. Some works in this list provide MRI studies of DNDs, which is not easy to find (the section 6 – MRI – in the manuscript is too short and do not cover any DND studies at all). 1) Enhancement of Singlet Oxygen Generation of Radachlorin® Conjugated with Polyvinylpyrrolidone and Nanodiamonds in Aqueous Media (10.1007/978-3-030-77371-7_10) – describing DNDs as carriers of photosensitizer for PDT; 2) New Photocatalytic Materials Based on Complexes of Nanodiamonds with Diphthalocyanines of Rare Earth Elements (10.1007/978-3-030-77371-7_7); 3) Suspensions of manganese-grafted nanodiamonds: Preparation, NMR, and MRI study (10.1016/j.diamond.2022.109591); 4) PVP-coated Gd-grafted nanodiamonds as a novel and potentially safer contrast agent for in vivo MRI (10.1002/mrm.28762); 5) Complexes of nanodiamonds with Gd-fullerenols for biomedicine (10.1080/1536383X.2021.1993443); 6) Diamond-based nanostructures with metal-organic molecules (10.1080/1539445X.2021.1992425); 7) Gadolinium-Complexed Carboxylated Nanodiamond Particles for Magnetic Resonance Imaging of the Lymphatic System (10.1021/acsanm.0c03165); 8) Manganese-grafted detonation nanodiamond, a novel potential MRI contrast agent (10.1016/j.diamond.2021.108590).
The Ehglish is good enough, some minor editing needed.
Author Response
Response to Reviewer 2 Comments
Referee #1:
Comments to the Author
The manuscript by Xinyue Wang et al. “Application of Nanodiamonds in Bioimaging: A review” presents a review of recent works on nanodiamonds applications in the field of bioimaging. The works on this subject are well documented. However, some questions arise along the reading, and some formulations need a clarification, especially in the introduction section.
- The introduction requires a brief description of NDs types and way of their producing. DNDs and HPHT NDs are only mentioned (line 58), and need some more insight of their nature – typical particle sizes, concentrations of fluorescent centers, applications, etc. Are their any other types of ND? The classification of NDs declared to be summarized (line 1698) in fact is not presented.
Authors’ response: We are very grateful for the questions raised by the reviewer. We have revised the Introduction section by adding a brief description of the types and synthesis methods of NDs, adding three types of NDs in addition to DNDs and HPHT NDs: high-energy ball-milled NDs, laser-ablated NDs, and chemical vapor deposited NDs. As well as a description of the synthesis methods commonly applied to DNDs and HPHT NDs, their properties and their potential for the application in biosciences is briefly described. We have added the corresponding references during the revision process and they will be listed below. We have made some modifications in the conclusion section according to the order of presentation in the introduction section.
Change#1: 1. Introduction
There are various types of NDs, and we can classify NDs using different criteria. Firstly, according to the spatial size, they can be classified into zero-dimensional ND single crystal particles, one-dimensional diamond nanorods, two-dimensional diamond nanosheets, three-dimensional ND polycrystalline particles and ND films. Secondly, they can be classified according to the synthesis methods of NDs, which can be divided into detonation nanodiamonds (DNDs), high pressure and high temperature nanodiamonds (HPHT NDs), high energy ball milling NDs, laser ablation NDs, and chemical vapor deposition NDs [6]. DNDs are synthesized by the carbon of the explosive system transformed into diamond by the high temperature and pressure generated during explosive mixtures detonated in the absence of oxygen. The synthesized NDs surface have many surface groups (including oxygen-containing functional groups or other functional groups) that change the surface potential of the DNDs, causing the DNDs to agglomerate, and thus the final result is generally a polymer of DNDs [6]. Generally, the size of DNDs after isolation and purification is 4-5 nm. DNDs are highly biocompatible and their numerous surface groups make them easy to surface modify which are beneficial for bioscience applications. HPHT NDs are synthesized by transforming graphite powder into NDs under quasi-hydrostatic pressure and high temperature. In recent studies, naphthalene (C10H8 (Chemapol)) and organosilicon compounds such as tetrakis (trimethylsilyl) silane (C12H36Si5 (Stream Chemicals)) were generally used as precursors and then the high-temperature and high-pressure methods were used to synthesize HPHT NDs [4,6]. In a new study, HPHT NDs were synthesized with a size of about 10 nm [7]. The HPHT NDs synthesized from organic matter using this method do not contain metal impurities and have good biocompatibility as a good material for bioimaging probes. The surface profiles of DNDs and HPHT NDs were characterized by Zeleneev et al. using confocal microscopy and atomic force microscopy (AFM) (These two types of microscopes we will describe in detail below) [4]. The experimental results show that, NV centers are mainly found in DNDs while SiV centers are mainly found in HPHT NDs. The average size of aggregates containing a single NV center is 60 nm. In the synthesis process of HPHT NDs, the vacuum deposition technique was adopted to avoid their agglomeration, and the SiV centers existed in separate HPHT NDs crystals, and the size of this HPHT NDs was 10 nm. For DNDs and HPHT NDs that contain both NV centers, due to the highly nonequilibrium (detonation synthesis) and near-equilibrium (high pressure and high temperature synthesis) synthesis conditions, the NV color center concentration of DNDs is lower than that of HPHT NDs, making the fluorescence performance of DNDs weaker than that of HPHT NDs [8]. Both types of NDs are able to exhibit high single-photon brightness, even at room temperature, making them good bioimaging probe materials for a wide range of applications in the life sciences. DNDs have greater advantages over HPHT NDs in bioimaging applications because of their smaller size, larger specific surface area and abundant surface groups, and therefore their high drug loading efficiency, allowing them to be used as nanocarriers to deliver various functional molecules such as contrast agents, proteins, nucleic acids and small molecule drugs [9].
- Zeleneev, A. I., Bolshedvorskii, S. V., Zhulikov, L. A., Sochenko, V. V., Rubinas, O. R., Sorokin, V. N., ... & Akimov, A. V. (2020, June). On studying the optical properties of NV/SiV color centers in ultrasmall nanodiamonds. In AIP Conference Proceedings (Vol. 2241, No. 1, p. 020039). AIP Publishing LLC.
- Qin, J. X., Yang, X. G., Lv, C. F., Li, Y. Z., Liu, K. K., Zang, J. H., ... & Shan, C. X. (2021). Nanodiamonds: Synthesis, properties, and applications in nanomedicine. Materials & Design, 210, 110091.
- Vervald, A. M., Burikov, S. A., Scherbakov, A. M., Kudryavtsev, O. S., Kalyagina, N. A., Vlasov, I. I., ... & Dolenko, T. A. (2020). Boron-doped nanodiamonds as anticancer agents: En route to hyperthermia/thermoablation therapy. ACS Biomaterials Science & Engineering, 6(8), 4446-4453.
- Kolarova, K., Bydzovska, I., Romanyuk, O., Shagieva, E., Ukraintsev, E., Kromka, A., ... & Stehlik, S. (2023). Hydrogenation of HPHT nanodiamonds and their nanoscale interaction with chitosan. Diamond and Related Materials, 134, 109754.
- Zhang, K., Guo, Q., Zhao, Q., Wang, F., Wang, H., Zhi, J., & Shan, C. (2021). Photosensitizer Functionalized NDs for Raman Imaging and Photodynamic Therapy of Cancer Cells. Langmuir, 37(14), 4308-4315.
Change#2: 9. Conclusion and perspective
In this review, we have summarized the classification of nanodiamonds, synthesis methods, optical defective color centers and nanodiamond surface modifications and the advantages of nanodiamonds as bioimaging probes compared to traditional fluorescent stains and quantum dots.
→In this review, we have summarized NDs optical defective color centers, the classification and synthesis methods of NDs (with a focus on DNDs and HPHT NDs) and the advantages of NDs as bioimaging probes compared to traditional fluorescent stains and quantum dots.
- The advantages of NDs for bioimaging (lines 74-83) is too brief and need an extended description. At least, two aspects are necessary to introduce. 1) Does the particle size of NDs, larger than many of fluorescent agent molecules, allow NDs to be effectively used as nanocarriers, penetrate through cellular membranes? 2) Are the NDs able to eliminate from the cells and tissues after therapy. To my knowledge, this task is not yet properly solved, and if NDs will accumulate in tissues without excretion, it may cause harmful consequences. This is not the subject of the manuscript, but some introduction into the problem needed to be described.
Authors’ response: Thank you very much for the questions raised by the reviewer. In the mentioned section, we provide an introduction to the transmembrane transport of NDs and elaborate on the in vivo long-term effects of NDs as drug transport carriers. To avoid the Introduction section too long, we have only briefly summarized the corresponding contents (in the next sections in the article), and the modified parts are listed below.
Change#1: 1. Introduction
Bioimaging facilitates the study of intracellular nanoparticle transport pathways, and the entry of NDs into cells is related to the size and shape of the NDs, the time that the NDs are co-cultured with the cells, and the material with which the NDs are complexed. In the next sections presented, the study of the size of NDs complexes confirms that NDs complexes with smaller sizes are more readily taken up by cells [12]. Different sizes of NDs are transported across membranes in different ways [13]. The time of NDs complexes infecting cells influences their entry into cell membranes [12]. The use of different materials for compounding with NDs or coating NDs affects the entry of NDs into cells [14]. Surface wrapping of NDs and aggregation of NDs can change their shape and thus affect their transmembrane transport status. In many ND-X therapeutic systems (NDs-based cancer treatment systems), NDs generally remain in the cell after transporting drug molecules and do not have a significant effect on the cell for a short period of time, but further studies are needed for a prolonged treatment process [15].
- Zhang, P., Yang, J., Li, W., Wang, W., Liu, C., Griffith, M., & Liu, W. (2011). Cationic polymer brush grafted-nanodiamond via atom transfer radical polymerization for enhanced gene delivery and bioimaging. Journal of Materials Chemistry, 21(21), 7755-7764.
- Perevedentseva, E., Ali, N., Karmenyan, A., Skovorodkin, I., Prunskaite-Hyyryläinen, R., Vainio, S., ... & Kinnunen, M. (2019). Optical studies of NDs-tissue interaction: skin penetration and localization. Materials, 12(22), 3762.
- Wu, Y., Alam, M. N. A., Balasubramanian, P., Winterwerber, P., Ermakova, A., Müller, M., ... & Weil, T. (2021). Fluorescent nanodiamond–nanogels for nanoscale sensing and photodynamic applications. Advanced NanoBiomed Research, 1(7), 2000101.
- Perevedentseva, E., Lin, Y. C., & Cheng, C. L. (2021). A review of recent advances in nanodiamond-mediated drug delivery in cancer. Expert Opinion on Drug Delivery, 18(3), 369-382.
- A misprint at the line 84: Fluorescence imaging.
Authors’ response: We thank the reviewer for the suggestions. In the revised version, we did spell check inside the article.
Change#1: Spell check
At the line 21: Florescence→Fluorescence
At the line 25: Florescence→Fluorescence
At the line 151: Florescence→Fluorescence
- The description at lines 85-87 is not correct: “For individual detonation 85 NDs due to its small size is prone to aggregation, and the fluorescence emitted from 86 smaller size NDs is weaker and cannot avoid the interference of tissue autofluorescence, 87 which affects the final imaging quality.” At least, can you provide a reference, where DNDs are prone to aggregation due to their small particle size? On the contrary – smaller size (typically ~ 5 nm) provides better stability, which cannot be achieved in the case of large micron and sub microscale particles. This is generally right for many types of particles, not only diamonds. The stable aqueous DND suspensions have been reported repeatedly (ex., doi: 10.1016/j.cplett.2016.06.010, 10.1039/d0cp05533f, 10.1134/S1027451012100151, 10.1021/nn100748k, 10.1016/j.carbon.2017.07.013, 10.1166/nnl.2011.1122). The paper 10.1021/nn100748k even declares that the size for the better photonic activity should be small, ~ 4 nm. DND can agglomerate and form clusters, because of other reasons, not small size – covalent bonds, coulombic and van der Waals interactions (10.1016/j.molliq.2022.118816, 10.1021/acs.jpcc.9b03175), which can be overcome after the process of deagglomeration. DND are prone to aggregation in saline buffers, including isotonic buffer, which can be the barrier for their medical use, but this problem has already been solved (10.1016/j.diamond.2018.05.012). In fact, the main reason for DNDs poor fluorescence is lower concentration of defects, compared to HPHT diamonds. This was reported in numerous works, one of the latest - 10.1016/j.diamond.2023.109754. The links are for your information, I don’t ask to cite them all.
Authors’ response: We appreciate the reviewer’s good suggestion. We corrected the errors that appeared and carefully read the literatures provided by the reviewer. In the preface of 2. Fluorescence imaging, the section on stable aqueous suspensions of DNDs has been added and the corresponding literatures have been cited. The advantages of small size DNDs are also described and references are cited. In the introduction section, the reasons why DNDs are prone to aggregation are introduced. In this section, the concentration of NV centers of both types of NDs with NV centers is introduced, and the corresponding references are cited. The changes made are listed below.
Change#1: 2. Fluorescence imaging
As mentioned above there are many functional groups on the surface of DNDs and these functional groups can have a significant effect on the properties of DNDs. In recent studies the aggregation stability of DNDs hydrosols has been related to the interaction of charged particles in the functional groups on the surface [16,17]. Dispersion of DNDs aggregates using a range of modalities contributes to the formation of stable aqueous suspensions of DNDs. For example, DNDs aggregates can be purified into small particles by powerful sonication and oxidative grinding in water [18], annealing the NDs aggregates powder (>100 nm) in hydrogen, the aggregates are broken down into their core (4 nm) particles, which are then dispersed into water by high power sonication and high speed centrifugation, producing monodisperse NDs colloids [19], sp3-sp2 rehybridization of carbon atoms on the surface of DNDs yielded 4-5 nm individual NDs particles aqueous solutions [20], rupturing DNDs agglomerates by deep purification and air annealing followed by centrifugation to produce DND hydrosols with high negative zeta potential [21]. Obtaining stable aqueous suspensions of DNDs of 4-5 nm is of great importance for surface modification of nanoparticles in biomedical applications. Smaller size DNDs can provide better stability and in one study, small size DNDs (~4nm) were able to obtain better photonic activity [19]. Surface functionalization of pristine NDs is an important component of fluorescence imaging, and Avdeev et al. reported two types of stabilization of DNDs in aqueous suspensions negative potential (-stabilization) and positive potential (+stabilization) [16]. Negative potential stabilization is achieved by annealing the DND powder in air at temperatures higher than 400°C, and positive potential stabilization is achieved by annealing the nanodiamonds in a hydrogen environment, followed by the binding of CH groups to the DND surface, creating a positively charged layer when interacting with water. Surface functionalization of NDs also involves wrapping a cationic polymer coating on the NDs surface can be used to localize and detect the polymeric nanoparticles by using the spectral properties of different polymers, or connecting the NDs with green fluorescent protein (GFP) to emit fluorescence by luciferase gene expression, which can track the cells containing the compounds.
- Avdeev, M. V., Tomchuk, O. V., Ivankov, O. I., Alexenskii, A. E., Dideikin, A. T., & Vul, A. Y. (2016). On the structure of concentrated detonation nanodiamond hydrosols with a positive ζ potential: Analysis of small-angle neutron scattering. Chemical Physics Letters, 658, 58-62.
- Knizhnik, A. A., Polynskaya, Y. G., Sinitsa, A. S., Kuznetsov, N. M., Belousov, S. I., Chvalun, S. N., & Potapkin, B. V. (2021). Analysis of structural organization and interaction mechanisms of detonation nanodiamond particles in hydrosols. Physical Chemistry Chemical Physics, 23(1), 674-682.
- Tomchuk, O. V., Avdeev, M. V., Aksenov, V. L., Garamus, V. M., Bulavin, L. A., Ivashevskaya, S. N., ... & Schreiber, J. (2012). Comparative structural characterization of the water dispersions of detonation nanodiamonds by small-angle neutron scattering. Journal of Surface Investigation. X-ray, Synchrotron and Neutron Techniques, 6, 821-824.
- Williams, O. A., Hees, J., Dieker, C., Jager, W., Kirste, L., & Nebel, C. E. (2010). Size-dependent reactivity of diamond nanoparticles. ACS nano, 4(8), 4824-4830.
- Dideikin, A. T., Aleksenskii, A. E., Baidakova, M. V., Brunkov, P. N., Brzhezinskaya, M., Davydov, V. Y., ... & Vul, A. Y. (2017). Rehybridization of carbon on facets of detonation diamond nanocrystals and forming hydrosols of individual particles. Carbon, 122, 737-745.
- Aleksenskiy, A. E., Eydelman, E. D., & Vul, A. Y. (2011). ’, Nanotechnol. Lett, 3, 68.
Change#2: 1. Introduction
The synthesized NDs surface have many surface groups (including oxygen-containing functional groups or other functional groups) that change the surface potential of the DNDs, causing the DNDs to agglomerate, and thus the final result is generally a polymer of DNDs [6]. Generally, the size of DNDs after isolation and purification is 4-5 nm.
- Qin, J. X., Yang, X. G., Lv, C. F., Li, Y. Z., Liu, K. K., Zang, J. H., ... & Shan, C. X. (2021). Nanodiamonds: Synthesis, properties, and applications in nanomedicine. Materials & Design, 210, 110091.
Change#3: 1. Introduction
For DNDs and HPHT NDs that contain both NV centers, due to the highly nonequilibrium (detonation synthesis) and near-equilibrium (high pressure and high temperature synthesis) synthesis conditions, the NV color center concentration of DNDs is lower than that of HPHT NDs, making the fluorescence performance of DNDs weaker than that of HPHT NDs [8].
- Kolarova, K., Bydzovska, I., Romanyuk, O., Shagieva, E., Ukraintsev, E., Kromka, A., ... & Stehlik, S. (2023). Hydrogenation of HPHT nanodiamonds and their nanoscale interaction with chitosan. Diamond and Related Materials, 134, 109754.
- “FNDs” are not introduced at the first time it appears, line 93. Are FNDs part of HPHT diamonds, or they are produced by another way? At the line 1482 the authors state that DNDs are the class of FNDs, which leaves some doubts.
Authors’ response: We appreciate the reviewer for the suggestions. FNDs refers to Fluorescent nanodiamonds. We added the markup when the FNDs first appeared. DNDs containing optically defective centers are able to emit fluoresce and can be considered as a type of FNDs, but the formulation is not rigorous at the mentioned position and we have made modifications.
Change#1:
At the line 188: Add “(Fluorescent nanodiamonds)”
Change#2:
At the line 1748: FNDs→NDs
- GNP is not introduced at the first mentioning, line 230.
Authors’ response: Thank you very much for the questions raised by the reviewer and we modified the mistake in the revised version.
Change#1:
At the line 357: Add “(nanodiamond–gold nanoparticle)”
- PEI is wrong reported as polyetherimide, line 271. In fact, the authors at the ref. 9 used poly(ethyleneimine).
Authors’ response: We thank the reviewer for the suggestions and we modified the mistake in the revised version.
Change#1:
At the line 402: polyetherimide→poly(ethyleneimine)
- At the line 529 some word is redundant (3D tomographic tomography).
Authors’ response: We thank the reviewer for the suggestions and we modified the mistake in the revised version.
Change#1:
At the line 703: Delete “and 3D tomographic tomography”
- At lines 600-602 the authors again describe wrong reason for DND agglomeration. The measured particle size 258 ± 60nm is surely the size of agglomerates. DND very rarely exist in the form of single particles, even in the stable suspensions, where they form chain clusters (10.1016/j.carbon.2016.12.007).
Authors’ response: We are very grateful for the questions raised by the reviewer. We have fixed the errors that occurred and listed the changes below.
Change#1:
At the line 781: Delete “due to the small size of the particles”
Change#2:
At the line 783: particles→polymers
- A probable misprint, line 997: “GNDs in NDs-GNP conjugates” should be “GNPs in NDs-GNP conjugates”.
Authors’ response: We appreciate the reviewer for the suggestions and we modified the mistake in the revised version.
Change#1:
At the line 1218: “GNDs in NDs-GNP conjugates” →“GNPs in NDs-GNP conjugates”
- I may suggest to follow some articles dealing with imaging of DNDs for possible consideration in the review to make it a little more complete, and to demonstrate that DNDs have some certain advances over HPHT diamonds, mainly due to DNDs small size and surface. Some works in this list provide MRI studies of DNDs, which is not easy to find (the section 6 – MRI – in the manuscript is too short and do not cover any DND studies at all). 1) Enhancement of Singlet Oxygen Generation of Radachlorin Conjugated with Polyvinylpyrrolidone and Nanodiamonds in Aqueous Media (10.1007/978-3-030-77371-7_10) – describing DNDs as carriers of photosensitizer for PDT; 2) New Photocatalytic Materials Based on Complexes of Nanodiamonds with Diphthalocyanines of Rare Earth Elements (10.1007/978-3-030-77371-7_7); 3) Suspensions of manganese-grafted nanodiamonds: Preparation, NMR, and MRI study (10.1016/j.diamond.2022.109591); 4) PVP-coated Gd-grafted nanodiamonds as a novel and potentially safer contrast agent for in vivo MRI (10.1002/mrm.28762); 5) Complexes of nanodiamonds with Gd-fullerenols for biomedicine (10.1080/1536383X.2021.1993443); 6) Diamond-based nanostructures with metal-organic molecules (10.1080/1539445X.2021.1992425); 7) Gadolinium-Complexed Carboxylated Nanodiamond Particles for Magnetic Resonance Imaging of the Lymphatic System (10.1021/acsanm.0c03165); 8) Manganese-grafted detonation nanodiamond, a novel potential MRI contrast agent (10.1016/j.diamond.2021.108590).
Authors’ response: We are very grateful for the questions raised by the reviewer. We have added DNDs imaging and added in the Introduction section that DNDs have advantages over HPHT NDs in the field of bioimaging. In section 2.2.1. Optical microscope, we added the use of DNDs as a photosensitizer carrier to make a photocatalyst to promote the production of singlet oxygen. In part 6. Magnetic resonance imaging, we added the use of DNDs as a carrier for MRI contrast agents to make MRI contrast agents with excellent properties and improve MRI imaging contrast. We have listed the modified parts below.
Change#1: 1. Introduction
DNDs have greater advantages over HPHT NDs in bioimaging applications because of their smaller size, larger specific surface area and abundant surface groups, and therefore their high drug loading efficiency, allowing them to be used as nanocarriers to deliver various functional molecules such as contrast agents, proteins, nucleic acids and small molecule drugs [9].
- Zhang, K., Guo, Q., Zhao, Q., Wang, F., Wang, H., Zhi, J., & Shan, C. (2021). Photosensitizer Functionalized NDs for Raman Imaging and Photodynamic Therapy of Cancer Cells. Langmuir, 37(14), 4308-4315.
Change#2: 2.2.1. Optical microscope
In recent studies, DNDs were used as carriers of PDT photosensitizers to form novel photocatalytic materials that facilitate further 1O2 production. Kulvelis et al. used the photosensitizer Radachlorin® (sodium salts of chlorin e6, chlorin p6, and purpurin 5) with polyvinylpyrrolidone (PVP) and DNDs to synthesize a ternary catalytic complex, using UV irradiating the complex to excite DNDs, which do not emit light and transfer energy to surrounding molecules thereby catalyzing the production of 1O2 [33]. Due to the chemical inertness of the DNDs in this new catalytic complex, they are resistant to 1O2 and can remain stable for a long time. Lebedev et al. developed a new photoactive catalyst synthesized from europium diphthalocyanine molecules dissolved in dimethylformamide and transferred into the aqueous dispersion of DNDs (~4.5 nm in size and positive ζ-potential of ~30-40 mV) to form diphthalocyanine phthalocyanine-NDs complexes [34]. This hybrid structure can be used as a catalyst to produce 1O2 in the surrounding medium (air, water, biological tissues) under light excitation at wavelengths ~600-700 nm [34].
- Kulvelis, Y. V., Lebedev, V. T., Yevlampieva, N. P., Cherechukin, D. S., & Yudina, E. B. (2022). Enhancement of Singlet Oxygen Generation of Radachlorin® Conjugated with Polyvinylpyrrolidone and Nanodiamonds in Aqueous Media. Green Photocatalytic Semiconductors: Recent Advances and Applications, 281-306.
- Lebedev, V. T., Tӧrӧk, G., Kulvelis, Y. V., Soroka, M. A., Ganzha, V. A., Orlova, V. A., ... & Shvidchenko, A. V. (2022). New Photocatalytic Materials Based on Complexes of Nanodiamonds with Diphthalocyanines of Rare Earth Elements. Green Photocatalytic Semiconductors: Recent Advances and Applications, 179-208.
Change#3: 6. Magnetic resonance imaging
MRI imaging requires a suitable contrast agent to obtain high quality MRI images. Conventional MRI contrast agents (gadolinium (Gd3+), manganese (Mn2+) and iron (Fe3+), etc.) improve the visibility of internal structures by shortening the relaxation time of aqueous hydrogen nuclei in biological tissues [61]. DNDs can be used as nanocarriers attached to metal particles to avoid the effects of potential toxicity of metal particles and to enable targeted MRI imaging. There are two main ways of attachment, one is surface modification of DNDs using groups containing the aforementioned particles. For example, the negatively charged fullerenol Gd@C82(OH)X (X∼30) was attached with positively charged DNDs to form a complex, which can form a stable chain structure, and due to complexation, the fullerenol has magnetic resonance properties that can improve the imaging contrast of MRI [62]. Lebedev et al. found in neutron scattering experiments that negatively charged Eu diphthalocyanines and DNDs with positive surface potential in aqueous media can be assembled into a complex that is stable in aqueous solution, and DNDs can improve the magnetic and optical activity of Eu diphthalocyanines [63]. Yano et al. used a pre-oxidation step to create abundant hydrophilic carboxyl groups on the surface of NDs, which are able to disperse the particles in aqueous solution. The carboxylated nanodiamond (CND) was then condensed with gadolinium chelate (Gd-DTPA) to obtain Gd-DTPA-CND complexes with a hydrodynamic diameter of about 4-5 nm, which can be used for high-resolution selective imaging of the lymphatic system [64]. Another approach is to graft metal ions (magnetic and luminescent) directly onto the surface of DNDs [63]. Panich et al. successfully prepared suspensions of Gd-DND at different concentrations and their suspensions showed high proton relaxivity. Such suspensions are promising as MRI contrast agents, which can shorten the spin-lattice (T1) and spin-spin (T2) relaxation times of water protons and increase the signal intensity of T1- and T2-weighted MRI images [65]. To maintain the stability of the Gd-DND complex in saline and to avoid its further coagulation in blood, the Gd-DND particles were coated with a polyvinylpyrrolidone (PVP) protective shell, and the obtained PVP-Gd-DND contrast agent provided higher T1-weighted hyperintense signal [61]. The high relaxivity and low toxicity of manganese compounds relative to gadolinium (III) ions allow their use as MRI contrast agents. Panich et al. recently prepared high-purity 4-5 nm DND powders, and the reaction of aqueous suspensions of DNDs with aqueous manganese sulfate solutions allows direct grafting of Mn2+ ions onto the surface of DNDs [65,66]. Similarly performing PVP coverage avoids possible coagulation of the particles in the blood. The interaction of ions with the electronic and nuclear spins of NDs accelerates the relaxation of the spin lattice, which facilitates the improvement of MRI imaging contrast [66].
- Panich, A. M., Salti, M., Prager, O., Swissa, E., Kulvelis, Y. V., Yudina, E. B., ... & Shames, A. I. (2021). PVP‐coated Gd‐grafted nanodiamonds as a novel and potentially safer contrast agent for in vivo MRI. Magnetic Resonance in Medicine, 86(2), 935-942.
- Lebedev, V. T., Kulvelis, Y. V., Peters, G. S., Bolshakova, O. I., Sarantseva, S. V., Popova, M. V., & Vul, A. Y. (2022). Complexes of nanodiamonds with Gd-fullerenols for biomedicine. Fullerenes, Nanotubes and Carbon Nanostructures, 30(1), 36-45.
- Lebedev, V. T., Török, G., Kulvelis, Y. V., Bolshkova, O. I., Yevlampieva, N. P., Soroka, M. A., ... & Garg, S.(2022). Diamond-based nanostructures with metal-organic molecules. Soft Materials, 20(sup1), S34-S43.
- Yano, K., Matsumoto, T., Okamoto, Y., Bito, K., Kurokawa, N., Hasebe, T., & Hotta, A. (2021). Gadolinium-complexed carboxylated nanodiamond particles for magnetic resonance imaging of the lymphatic system. ACS Applied Nano Materials, 4(2), 1702-1711.
- Panich, A. M., Salti, M., Aleksenskii, A. E., Kulvelis, Y. V., Chizhikova, A., Vul, A. Y., & Shames, A. I. (2023). Suspensions of manganese-grafted nanodiamonds: Preparation, NMR, and MRI study. Diamond and Related Materials, 131, 109591.
- Panich, A. M., Shames, A. I., Aleksenskii, A. E., Yudina, E. B., & Vul, A. Y. (2021). Manganese-grafted detonation nanodiamond, a novel potential MRI contrast agent. Diamond and Related Materials, 119, 108590.
Finally, we thank the editor and reviewers very much again for everything done for us. The insightful comments improved the quality of this manuscript. We hope the revised manuscript is acceptable for publication.
Thank you for your consideration.
Yours Sincerely,
Dr. Dandan Sang
Reviewer 3 Report
Athors have presented a very well framed review article deciphering the potential of nanodiamonds in Bioimaging. Title is good however author can reframe the title to enhance the readability of the review article. The abstract section is very well explained. The introduction part is fair enough to summarise the zest of the review article. All the sections are very well elaborated and a good number of figures are used in this review. Are these figures from the authors' research work? If yes then it is ok otherwise please take the copyright. The conclusion section has explained the future perspective of the review article. Altogether, I am highly impressed with the review article and recommend it for publication. However, I would request the author to do a spelling check throughout the manuscript. Here are a few suggestions:
1. Please check the spelling of florescence in section 2 as the author has sometimes used fluorescence. Kindly check throughout the manuscript.
2. Enhance the picture quality of Figure 1
Author Response
Response to Reviewer 3 Comments
Referee #1:
Comments to the Author
Authors have presented a very well framed review article deciphering the potential of nanodiamonds in Bioimaging. Title is good however author can reframe the title to enhance the readability of the review article. The abstract section is very well explained. The introduction part is fair enough to summarize the zest of the review article. All the sections are very well elaborated and a good number of figures are used in this review. Are these figures from the authors' research work? If yes then it is ok otherwise please take the copyright. The conclusion section has explained the future perspective of the review article. Altogether, I am highly impressed with the review article and recommend it for publication. However, I would request the author to do a spelling check throughout the manuscript. Here are a few suggestions:
- Please check the spelling of florescence in section 2 as the author has sometimes used fluorescence. Kindly check throughout the manuscript.
Authors’ response: We appreciate the reviewer’s good suggestion. In the revised version, we modified the title, and did spell check inside the article. The figures that appear in the article are from the references, and we have obtained the copyright of the figures and marked the copyright information in the figure notes section. We list the changes below.
Change#1: The title
Application of Nanodiamonds in Bioimaging: A review→Multiple bioimaging applications based on the excellent properties of nanodiamond: A review
Change#2: Spell check
At the line 21: Florescence→Fluorescence
At the line 25: Florescence→Fluorescence
At the line 151: Florescence→Fluorescence
- Enhance the picture quality of Figure 1
Authors’ response: We are very grateful for the questions raised by the reviewer. We have reworked Figure 1 and improved the picture quality.
Change#1: Figure 1
Figure 1. (a) Photoluminescence emission spectra of NV− and NV0 centers. The star marks the zero-phonon line of NV− (637 nm) and the zero-phonon line of NV0 (575 nm). The inset shows the structure of the NV center. [Reprinted with permission from Ref. [13]. Copyright 2010, American Physical Society]. (b) PL spectra of Microdiamant™ polycrystalline diamond particles with various submicron fractions: blue—DP 0–0.05 (mean size 25 nm), green—DP 0–0.2 (mean size 100 nm), red—DP 0–0.35 (mean size 175 nm). Under laser with excitation wavelength λ=488 nm. The prominent peak at 738 nm is the zero-phonon line of negatively charged SiV− centres which marked by the vertical dashed line, and it can be observed in all polycrystalline diamond fractions. The spectra are specially normalized for PL intensity at λ =590 nm in order to produce better comparison. In the figure shows normalising coefficients. [Reprinted with permission from Ref. [11]. Copyright 2020, Springer US]. (c) PL emission spectra of NDs obtained under 532 nm (1) and 325 nm (2) laser excitation with peaks showing fluorescence emission from 1.4 eV nickel-associated vacancy centers (d), (1) PL spectra of NDs of different sizes obtained using single-photon excitation of 325 nm laser at room temperature, (i) 100 nm, (ii) 500 nm, and (iii) 25 μm. (2) PL spectra of NDs of different sizes obtained using two-photon excitation with a 760 nm laser at room temperature, with the sizes of NDs marked on the upper right. [Reprinted with permission from Ref. [5]. Copyright 2018, Nizhny Novgorod State Medical Academy of the Ministry of Health of the Russian Federation]
Finally, we thank the editor and reviewers very much again for everything done for us. The insightful comments improved the quality of this manuscript. We hope the revised manuscript is acceptable for publication.
Thank you for your consideration.
Yours Sincerely,
Dr. Dandan Sang
Round 2
Reviewer 1 Report
The effort of the authors to write and edit this enormous manuscript is evident but not very effective. The manuscript is still very long, even longer than before the revision. That makes it extremely difficult to read. The sentences are still very long and confusing. It is necessary to shorten and improve the text in order to be published.
Language requires moderate changes.
Author Response
Referee #1:
Comments to the Author
The effort of the authors to write and edit this enormous manuscript is evident but not very effective. The manuscript is still very long, even longer than before the revision. That makes it extremely difficult to read. The sentences are still very long and confusing. It is necessary to shorten and improve the text in order to be published.
Authors’ response: Thank you very much for the questions raised by the reviewer. We deeply aware that the excessive length of the article makes it very difficult for readers to read. We have fine-tuned the whole text sentence by sentence, simplified the long sentences and cut out the less important conclusions. We also deleted the redundant pictures and simplified the corresponding conclusions. We rechecked the structure of the article and merged similar conclusions to make the structure of the article clearer. Since it is difficult to list all the deletions, we only list the deleted images.
Change#1: 2.2.1. Optical microscope
Delete Figure 2.
Figure 2. Optical images of A549 cells obtained after 24 hours of incubation in culture medium containing FND-Au. (a) Bright field optical image, (b) green fluorescence channel image (excitation wavelength λ=561 nm, fluorescence emission wavelength λ=575-718 nm), (c) composite image of Figure (a) and Figure (b). (d) green fluorescence channel image (excitation wavelength λ=561 nm, fluorescence emission wavelength λ=575-718 nm), (e) gray scattering channel image (excitation wavelength λ=458 nm, scattering wavelength λ=450-470 nm), (f) composite image of Figure (d) and Figure (e). Scale bar: 20 μm. [Reprinted with permission from Ref. [31]. Copyright 2016, American Chemical Society].
Delete Figure 4
Figure 4. The expression of green fluorescent protein was detected after culturing IC-21 cells in culture medium containing both FND-PEI and X-tremeGENE HP transfection reagent for 48 hours. The assay showed that both infectious reagents complexed with the pGFP expression plasmid. FND-PEI: images obtained by culturing cells in culture medium containing 25 micrograms per milliliter of FND-PEI; GFP: fluorescent images obtained by transfecting cells with X-tremeGENE HP transfection reagent and 2 micrograms per milliliter of pGFP; FND-PEI-GFP: fluorescence images obtained after transfecting cells with 25 μg per ml of FND-PEI and 2 μg per ml of pGFP. From left to right, DAPI staining of the nucleus; GFP fluorescence signal; fluorescence signal of the NV center; merged fluorescence image. Arrows point to cells positively transfected with NDs or plasmid DNA. [Reprinted with permission from Ref. [22]. Copyright 2016, Royal Society of Chemistry]
Change#2: 2.2.2. Electron microscope
Delete Figure 6.
Figure 6. FND−Au cell distribution and cell slice observed by TEM using the high-pressure freezing procedure. (a) Single FND−Au inside the cellular endosome. (b,c) FND−Au distributed in endosomes and cytosol. (d) Disrupted endosome membrane by the sharp FND edges. (e) Amplification of (d) [Reprinted with permission from Ref. [31]. Copyright 2016, American Chemical Society].
Delete Figure 7.
Figure 7. High-resolution TEM images of selected cubic diamond polycrystals extracted from the DP 0–0.05 powder (a, b) and typical images of the simple twinning boundaries, of several nanometres in length, found occasionally in the sample (c, d). Panels (a, b): Scale bar: 4 nm. Different crystallites are highlighted in different colours in (a). Arrows in panels (c) and (d) mark the selected clearly distinguishable twinning boundaries [Reprinted with permission from Ref. [25]. Copyright 2020, Springer US].
Delete Figure 8.
Figure 8. Imaging of individual FNDs in cellular submicrostructures. (a) Bright-field TEM of a single FND, (b) Dark-field EFTEM of a single FND; (e) Bright-field TEM of an intra-mitochondrial FND, (f) Dark-field EFTEM of an intra-mitochondrial FND. (c,g) Virtual sections of tomograms, (d,h) Segmentation of tomograms of single FND uptake and intra-mitochondrial localization are given, respectively. Scale bar: 200 nm [Reprinted with permission from Ref. [11]. Copyright 2019, American Chemical Society].
Delete Figure 9.
Figure 9. (a,b) SEM images of ND-SF spheres at 25 K and 250 K magnification, respectively. (c) After 7 days of incubation, the spheres degraded and only the NDs remained at 100 K magnification. (d) Scanning of NDs on the substrate at 100k magnification [Reprinted with permission from Ref. [37]. Copyright 2015, American Chemical Society].
Delete Figure 10.
Figure 10. SEM images of different sizes of NDs (100ND, 50ND, DND) on silicon substrates [Reprinted with permission from Ref. [13]. Copyright 2019, MDPI (Basel, Switzerland)].
Change#3: 2.3. Fluorescence lifetime imaging
Delete Figure 13.
Figure 13. Au-NDs and NDs were injected into Hela cells, respectively. (a, b) and (c, d) plots show the imaging analysis of both nanoparticles. (a, c) and (b, d) show the If and τf plots of NV centers, respectively. Scale bars: (a, c) 10 μm and (b, d) 2 μm [Reprinted with permission from Ref. [38]. Copyright 2017, American Chemical Society]. (e) Time decay of 100 ND fluorescence measured at two-photon femtosecond excitation (under a pulse laser at 800 nm); inset-FLIM of 100 ND particles positioned on an Si substrate [Reprinted with permission from Ref. [13]. Copyright 2019, MDPI (Basel, Switzerland)].
Delete Figure 15.
Figure 15. In vitro wide-field fluorescence imaging of FND-labeled cells in blood. Plots (a), (b) show the control group without time gating treatment and the experimental group with time gating treatment, respectively. A 1003 oil immersion objective was used for imaging during the acquisition of the wide-field fluorescence images, and the exposure times for the (a), (b) plots were 0.1 s and 0.3 s, respectively; FND-treated cells were covered with chicken breast in vitro to simulate in vivo imaging. (c), (d) Both figures are covered with a 0.1 mm thick layer of chicken breast on a glass slide of FND-labeled HeLa cells. The (c) plot is untreated by time gating technique and the (d) plot is treated by time gating technique. (e) Graph shows the fluorescence images obtained for the same FND-labeled HeLa cells without the chicken breast covering the glass slide and after the time gating treatment. (f) are the corresponding fluorescence intensity profiles obtained along (c), (d) and (e) plots, respectively [Reprinted with permission from Ref. [45]. Copyright 2014, Springer Nature].
Change#4: 2.4. Super-resolution optical imaging
Delete Figure 18.
Figure 18. (a) Time series microscopy image of FND with dashed circles showing particles with fluorescence emission fluctuations. Scale bar: 10 µm. (b) From left to right in the figure, applying a localization algorithm to detect a single nanoparticle on the sequence image, an example of super-resolution image with intensity profile over time and NDs signal intensity curve. Two fitted Gaussian curves with a distance of 187 nm appear in the figure, indicating that the distance between the two particles exceeds the diffraction limit (i.e., λ/2~325 nm) [Reprinted with permission from Ref. [30]. Copyright 2022, Multidisciplinary Digital Publishing Institute (MDPI)].
Change#5: 3. Raman imaging
Delete Figure 21.
Figure 21. Imaging detection of HeLa cells based on dual-color Raman imaging mode. (a) (a1) Bright field optical image; (a2) Raman image of HeLa cells; (a3) Raman image of NDs-Ce6 distribution in HeLa cells; (a4) Overlay of (a1, a2, a3) images. (b) Raman spectral lines corresponding to the white dot part of the (a, a1) image. Scale bar: 8 μm [Reprinted with permission from Ref. [9]. Copyright 2021, American Chemical Society].
Change#6: 4. X-ray imaging
Delete Figure 22.
Figure 22. (a) shows XRD images of raw, non-irradiated, and ion-implanted samples (ion injection time is on the left side of the image, and the number on the right side indicates the particle size) from left to right with three peaks corresponding to the (111), (220), and (311) planes of diamond (Reprinted with permission from Ref. [40]. Copyright 2021, American Chemical Society). (b) indicates the XRD patterns of cND, Co3O4 and cND-Co3O4 nanocomposites (Reprinted with permission from Ref. [54]. Copyright 2017, Elsevier).
Delete Figure 23.
Figure 23. Deconvoluted C 1s XPS spectra of DNDs. (a) Irradiation time of 7.5 min. (b) Irradiation time of 20 min. The inset section shows the sample photos [Reprinted with permission from Ref. [40]. Copyright 2021, American Chemical Society].
Change#7: 6. Magnetic resonance imaging
Delete Figure 27.
Figure 27. (a) Examination of the main first-order EPR signals of D19 diamond crystal at different microwave powers. The triple HFS structure of the EPR signal of the P1 center (substituted nitrogen) can be clearly discerned in the figure. (b) The upper and lower curves show the decomposition of the integrated EPR signal of the D19 diamond crystal into components related to the two sets of spins. The triple structure of curve 3 is consistent with the EPR spectrum of only the P1 center. (c) indicates the corresponding saturation trend curve of the peak intensity of this EPR signal about the square root of the microwave power (peak intensity below 200 mW). The dashed line is the I pp of DP 0-0.05 polycrystal versus (PMW)1/2 for reference. Four experimental points at PMW = 0.5, 1, 2, and 4 mW were used to plot. (Microwave frequency ν = 9.44 GHz) [Reprinted with permission from Ref. [25]. Copyright 2020, Springer US].
Change#8: 7. Cathodoluminescence imaging
Delete Figure 29.
Figure 29. CL plots of three semiconductor NPs obtained with the help of STEM. The peak of the spectrum of NDs containing NV centers appears at 620 nm (red), the peak of the spectrum of LuAG:Ce nanophosphates appears at 510 nm (green), and the peak of the spectrum of NDs with "band A" defects appears at 420 nm (blue). The normalized NP-CL spectra can be used as a basis for the selection of optical interference filters, with their passbands represented by colored rectangles, which can be used for color Cathodoluminescence imaging of each spectrally distinguishable particulate matter [Reprinted with permission from Ref. [67]. Copyright 2012, Springer Nature].
Change#9: 8. Optical coherence tomography imaging
Delete Figure 31.
Figure 31. Two-dimensional OCT images obtained from the detection of black skin samples. (a) indicates a control sample with normal structure, the yellow dashed line indicates the basal layer, and the red arrows indicate the separated dermis and hair follicles. (b) indicates skin samples treated with 100ND. (c) indicates skin samples treated with DNDs. White arrows indicate the stratum corneum with higher reflectance. Scale bar: 300 μm [Reprinted with permission from Ref. [13]. Copyright 2019, MDPI (Basel, Switzerland)].
Finally, we thank the editor and reviewers very much again for everything done for us. The insightful comments improved the quality of this manuscript. We hope the revised manuscript is acceptable for publication.
Thank you for your consideration.
Yours Sincerely,
Dr. Dandan Sang

Reviewer 2 Report
At the lines 678, 681 the authors use the term "polymers" to describe the associates of ND particles. Polymer is not a correct term. These particle associates are usually called clusters, aggregates or agglutinates in the case of DNDs.
Similar remark seem to be for "FND-Au copolymers" at the line 325. In fact, in the ref. 31 FND-Au are called hybrids, nanoparticles, hybrid nanoparticles, but not polymers.
The paper may be accepted after fixing these mistakes.
Author Response
Referee #1:
Comments to the Author
At the lines 678, 681 the authors use the term "polymers" to describe the associates of ND particles. Polymer is not a correct term. These particle associates are usually called clusters, aggregates or agglutinates in the case of DNDs.
Similar remark seem to be for "FND-Au copolymers" at the line 325. In fact, in the ref. 31 FND-Au are called hybrids, nanoparticles, hybrid nanoparticles, but not polymers.
The paper may be accepted after fixing these mistakes.
Authors’ response: We are very grateful for the questions raised by the reviewer. The mentioned "polymers" in lines 678 and 681 has been deleted, as the article is too long, only the main conclusion was kept in this section. For "FND-Au copolymers" in line 325, we searched the original literature and modified it to "FND-Au coaggregates" according to the original literature. After checking the full text, we found that the problem also exists in the Introduction section and made changes. The changes are listed below.
Change#1: 1. Introduction
At line 71: “a polymer of DNDs” → “aggregated NDs”
Change#2: 2.2.1. Optical microscope
At line 253: “FND-Au copolymers” →“FND-Au coaggregates”
Finally, we thank the editor and reviewers very much again for everything done for us. The insightful comments improved the quality of this manuscript. We hope the revised manuscript is acceptable for publication.
Thank you for your consideration.
Yours Sincerely,
Dr. Dandan Sang
